# GENERALIZATION OF NOISY SGD UNDER ISOPERIMETRY

## ABSTRACT

We study the generalization of iterative noisy gradient schemes on smooth non-convex losses. Formally, we establish time-independent information theoretic generalization bounds for Stochastic Gradient Langevin Dynamics (SGLD) that do not diverge as the iteration count increases. Our bounds are obtained through a stability argument: we analyze the difference between two SGLD sequences ran in parallel on two datasets sampled from the same distribution. Our result only requires an isoperimetric inequality to hold, which is merely a restriction on the tails of the loss. We relax the assumptions of prior work to establish that the iterates stay within a bounded KL divergence from each other. Under an additional dissipativity assumption, we show that the stronger Renyi divergence also stays bounded by establishing a uniform log-Sobolev constant of the iterates. Without dissipativity, we sidestep the need for local log-Sobolev inequalities and instead exploit the regularizing properties of Gaussian convolution. These techniques allow us to show that strong convexity is not necessary for finite stability bounds and thus for finite generalization and differential privacy bounds.

## 1 INTRODUCTION

Learning algorithms whose outputs are not highly sensitive to the specifics of their training data are likely to generalize well. This is the intuitive idea that undergirds the framework of information-theoretic generalization. The seminal contributions of Russo & Zou (2016) and Xu & Raginsky (2017) establish that the expected generalization gap of an algorithm $\mathcal{A}$ can be controlled by the amount of information the algorithm extracts from its training dataset $\mathcal{D}$ of size $n$. Formally, they show under reasonable assumptions that

$$|\text{generalization gap}(\mathcal{A}, \mathcal{D})| \leq \mathcal{O}\left(\sqrt{\frac{I(\mathcal{A}(\mathcal{D}), \mathcal{D})}{n}}\right)$$

where the input-output mutual information $I(\mathcal{A}(\mathcal{D}), \mathcal{D})$ measures the dependence of the algorithm's output on the observed samples $\mathcal{D}$. Unlike uniform-convergence-based bounds, information-theoretic bounds depend on the algorithm and the data distribution. This makes them well suited to assess the performance of models whose complexity renders vacuous all classic uniform-convergence bounds. More importantly, they align with the practical observations that generalization is data distribution dependent, as observed in experiments contrasting random versus real labels Zhang et al. (2021).

For them to be useful however a major difficulty remains in controlling this input-output mutual information for specific algorithms. Of particular interest for machine learning are algorithms obtained as iterative noisy gradient schemes. A standard template that these possess is the following. For a given dataset $\mathcal{D}$, a set of weights are randomly initialized $X_0 \in \mathbb{R}^d$ then updated following the recursion

$$X_{k+1} = \text{Noise}(\text{Gradient step}(X_k, \mathcal{D})) \tag{1}$$

The information-theoretic analysis of these algorithms was initiated in Pensia et al. (2018) where the gradient step is assumed to be bounded. Their results, as well as several follow-ups, are derived by viewing the algorithm as composition of individual steps analyzed separately. As a consequence, after $T$ iterations of the algorithm, the bounds obtained on $I(X_T, \mathcal{D})$ scale as $O(T)$ or $O(\sqrt{T})$

for fixed or non-vanishing step-sizes. The introduction of well-chosen analytical tools like data-dependent priors Haghifam et al. (2020), or the clever refinements of Bu et al. (2020) improve the bounds but still fall short of improving the time dependence.

The step-wise analysis leads to vacuous generalization bounds as iterations increase even if the recursion in equation 1 converges to a limit. Even more curious is the fact that this limit approximates the Gibbs distribution $e^{-F_n}$, where $F_n$ is the optimized training loss, which has been shown to achieve a finite information-theoretic bound Xu & Raginsky (2017); Pensia et al. (2018). We are faced with the strange situation where the iterates have an exploding bound but converge close to a distribution that has a finite one. Our aim in this work amend this strange gap to answer the following question:

*Do noisy iterative schemes in non-convex settings admit generalization bounds that go to zero as $n \to \infty$ without becoming vacuous as the number of iterations increases?*

Our main motivation for tackling this question is to understand whether *early-stopping* is necessary for generalization. If generalization gaps indeed diverge as iterations increase, then long training runs with non-vanishing step sizes are *proscribed*, by theory. If the divergence is merely the result of a loose analysis, then we should be able to establish better bounds that are more faithful to practice, where long training runs are common Power et al. (2022); Nakkiran et al. (2019).

We thus want to establish properties of noisy iterative schemes in their most realistic setting, i.e., when they are run for thousands of iterations with non-vanishing stepsizes. The algorithm we study as a representative of such schemes is noisy SGD, or Stochastic Gradient Langevin Dynamics (SGLD)Welling & Teh (2011). To study its generalization, we show that characterizing its stability is sufficient. Informally speaking, if SGLD outputs weights that are close (in a well-defined sense) when ran on two, different, independently-sampled datasets then it must not be overfitting. The difference between the outputs of SGLD measured using the KL and a stronger Rényi divergence relates to the expected generalization of the algorithm. Rényi stability goes further and also relates to differential privacy, a notion closely related to generalization. Through these techniques, we will show that noisy iterative schemes can have finite generalization and privacy bounds in unbounded non-convex settings, even when run for a large number of iterations. We make the following precise contributions.

**Contributions:**

- Under a structural assumption on the optimized loss, namely dissipativity, we show that uniform-in-time bounds can be established for both generalization and $(\epsilon, \delta)$-differential privacy of noisy SGD. Our bound only involves stability-related quantities and does not rely on ergodicity. We thus improve over the prior work in this setting namely Farghly & Rebeschini (2021); Futami & Fujisawa (2024); Zhu et al. (2024) who either rely on ergodicity (thus involving non-stability related constants) or obtain bounds that do not decay to zero as $n \to \infty$.

- To achieve our result, we resolve in passing an open question of Vempala & Wibisono (2019) on the isoperimetric properties of the biased limit of discrete Langevin iterates. We show that under dissipativity, all the iterates verify a uniform log-Sobolev inequality, a result which, to date, was only shown under strong convexity.

- As dissipativity is a crude assumption used control the log-Sobolev constant that often introduces constants exponential in dimension, we establish a secondary result that removes the dissipativity assumption but exploits ergodicity. Our bound in this case is polynomial in dimension and in the Gibbs' distribution's log-Sobolev constant. Unlike the analysis of Futami & Fujisawa (2024) who use dissipativity and rely on an involved extension of the parametrix method to unbounded drifts Bally & Kohatsu-Higa (2015), our analysis relaxes dissipativity and only exploits the regularizing properties of Gaussian convolution.

## 2 SETUP

In this section, we set up the notation, the definitions and the quantities we will analyze in the rest of the paper.

## 2.1 NOTATION

A central object in our analysis will be probability distributions over $\mathbb{R}^d$. All considered distributions are absolutely continuous with respect to the Lebesgue measure and admit a continuously differentiable density. For a distribution $a$ over $\mathbb{R}^d$, we will conflate the distribution and its density and denote $\nabla \log a$ as the gradient of its log-density and $\mathbb{E}_a$ the expectation under $a$. For $q > 0$, the $q$-Rényi divergence of $a$ with respect to $b$ is given by $D_q(a\|b) = \frac{1}{q-1} \log\left(\mathbb{E}_b\left[\left(\frac{a}{b}\right)^q\right]\right)$. It is a generalization of the Kullback-Leibler divergence (or relative entropy) which is recovered by taking the limit $D_{KL}(a\|b) := D(a\|b) := \lim_{q\to 1} D_q(a\|b)$. For random variables $X, Y$ with distribution $a, b$, we will denote, with a slight abuse of notation $D_q(X\|Y) := D_q(a\|b)$.

## 2.2 EMPIRICAL LOSS MINIMIZATION WITH SGLD

In supervised learning, the aim is to minimize a population risk of the form $F(x) := \mathbb{E}_\nu[f(x, Z)]$ with respect to $x \in \mathbb{R}^d$, where $Z \sim \nu$ is some unknown probability measure over some set $\mathcal{Z}$. Given access to a dataset $\mathcal{D}$ of $n$ independent, identically distributed samples $\mathcal{D} = Z_1, \ldots, Z_n$ from $\nu$, we optimize the empirical approximation $F_n$ given by $F_n(x, \mathcal{D}) = \frac{1}{n} \sum_{i=1}^n f(x, Z_i)$. We perform this minimization by assuming access to unbiased estimates of the gradient of $\nabla F_n$ of through minibatches of the form $g(x, B) = \frac{1}{|B|} \sum_{i \in B} \nabla f(x, Z_i)$ where $B = i_1, \ldots, i_b \subset \{1, \ldots, n\}$ are i.i.d uniform indices chosen from $[n]$. We have that $\mathbb{E}_B[g(x, B)] = \nabla F_n(x, D)$. With this in hand, the recursion we study to minimize $F_n$ is the following. An initial set of weights $X_0 \in \mathbb{R}^d$ is randomly sampled, then updated as follows

$$X_{k+1} = X_k - \eta g(X_k, B_k) + \sqrt{\frac{2\eta}{\beta}} N_{k+1} \qquad \text{(SGLD)}$$

where $\eta > 0$ is the stepsize, $(B_k)_k$ is a (conditionally) independent sequences of batches, $(N_k)_k$ are independent $\mathcal{N}(0, I)$ random variables, and $\beta > 0$ is a temperature parameter that scales the amount of noise injected. We refer to this recursion as *noisy* SGD as it corresponds to the SGD iterates with additional Gaussian noise added on top.

## 2.3 INFORMATION THEORETIC GENERALIZATION

A quantity of interest is how well optimizing the empirical proxy $F_n$ transfers to $F$. The SGLD algorithm ran for $k$ iterations is a randomized algorithm that outputs a random variable $X_k$ with distribution $\mathbb{P}_{X_k|\mathcal{D}}$ and the gap

$$\text{gen}(\mathbb{P}_{X_k|\mathcal{D}}, \nu) := |\mathbb{E}_{D,X_k}[F(X_k) - F_n(X_k, \mathcal{D})]|$$

where the expectations is taken with respect to $\mathcal{D} \sim \nu^{\otimes n}$ and $X_k \sim \mathbb{P}_{X_k|\mathcal{D}}$ measures how well the algorithm *generalizes* through the discrepancy between the loss achieved on the empirical loss versus the population one. Using a change of measure argument, Xu & Raginsky (2017) show that the following assumption is sufficient to control the generalization gap.

**Assumption 1** (Sub-Gaussian loss). *There exists $c_{\text{sg}} > 0$ such that for any $w \in \mathbb{R}^d$, the random variable $f(w, Z)$ is sub-Gaussian with variance proxy $c$ when $Z \sim \nu$* [1].

For losses verifying the assumption above, Xu & Raginsky (2017)'s work shows that KL stability of the algorithm controls the expected generalization gap (see Appendix G for a short proof).

**Lemma 2** (From KL-stability to generalization). *Let $\mathcal{D}, \mathcal{D}'$ be two independent samples from $\nu^{\otimes n}$. It holds under Assumption 1 that*

$$\text{gen}(\mathbb{P}_{X_k|\mathcal{D}}, \nu) \leq \sqrt{\frac{2c_{\text{sg}}\mathbb{E}_{\mathcal{D},\mathcal{D}'}\left[D_{KL}\left(P_{X_k|\mathcal{D}}\|P_{X_k|\mathcal{D}'}\right)\right]}{n}}.$$

To control the generalization gap, it therefore suffices to control distance between two sets of SGLD iterates ran on two datasets $\mathcal{D}$ and $\mathcal{D}'$ (see figure 1). Formally, by considering the KL divergence between iterates of

$$X_{k+1} = X_k - \eta g(X_k, B_k) + \sqrt{\frac{2\eta}{\beta}} N_{k+1} \ \text{ and } \ X'_{k+1} = X'_k - \eta g'(X'_k, B'_k) + \sqrt{\frac{2\eta}{\beta}} N'_{k+1} \quad (2)$$

---

[1] A random variable $X$ is sub-Gaussian with proxy $c$ if for $\lambda \in \mathbb{R}$, $\log \mathbb{E}[\exp(\lambda(X - \mathbb{E}(X)))] \leq \lambda^2 c^2/2$

which are two sequences SGLD iterates on two different independent datasets $\mathcal{D}$ and $\mathcal{D}'$, we can obtain upper bounds on the generalization gap.

## 2.4 DIFFERENTIAL PRIVACY

A closely related notion to generalization is differential privacy. For $\epsilon, \delta > 0$, $(\epsilon, \delta)$-Differential privacy Dwork (2008) is a standardized formalization of the notion of *privacy*. A useful interpretation of it is given by Wasserman & Zhou (2010) who show that an $(\epsilon, \delta)$-differential privacy guarantee on an algorithm $\mathbb{P}_{X|\mathcal{D}}$ outputting weights $X$ given a dataset $\mathcal{D}$, equates to a guarantee that no statistical test (or null-hypothesis test) on the output can reliably determine if a specific data point was part of the training set $\mathcal{D}$. The false-positive and false-negative rates of any such test will be controlled by $\epsilon$ and $\delta$. Algorithms with small $\epsilon$ and $\delta$ are those for which no powerful test exists. Remarkably, as shown below, stability in terms of the stronger Rényi divergence implies that an algorithm is $(\epsilon, \delta)$-differential private.

**Lemma 3** (From Rényi stability to $(\epsilon, \delta)$-DP (Thm.21 Balle et al. (2020))). *Let $P_{X_k|\mathcal{D}}$ be a randomized algorithm outputting weights given a dataset $\mathcal{D}$. Let $q > 1, \epsilon > 0$. If $D_q\left(P_{X_k|\mathcal{D}} \| P_{X_k|\mathcal{D}'}\right) \leq \epsilon$ for $\mathcal{D}, \mathcal{D}'$ adjacent datasets, then $P_{X_k|\mathcal{D}}$ is $(\epsilon + \frac{\log 1/\delta - \log(q)}{q-1} + \log \frac{q-1}{q}, \delta)$-differentially private for any $\delta > 0$.*

The study of the privacy properties of noisy iterative schemes appears in Minami et al. (2016). A comprehensive treatment from a perspective of *privacy amplification by iteration* in convex setting is provided in Feldman et al. (2018) using Rényi differential privacy Mironov (2017). Fundamentally, the technical problem of showing stability is identical for generalization and privacy, which is why we mention differential privacy here. The analysis of Ganesh & Talwar (2020); Chourasia et al. (2021); Ye & Shokri (2022) establishes time-independent privacy bound for strongly convex settings with deterministic and stochastic gradients. Going *beyond convexity* as noted in Ganesh et al. (2023) remained an open question.

## 2.5 ISOPERIMETRY

In our work, we relax the strong-convexity requirements to assumptions of *dissipativity* and *isoperimetry*. The inequality below is referred to as *an isoperimetric inequality* since it implies Gaussian (or uniform on the unit sphere) like concentration properties on the distribution Gozlan (2009).

**Definition 4** (LSI). *A distribution $b$ is said to verify the log-Sobolev inequality (LSI) with constant $c_b$ if for any $a \ll b$,*

$$\mathrm{D}_{KL}\left(a \| b\right) \leq \frac{c_b}{2} \mathbb{E}_a \left[\|\nabla \log a - \nabla \log b\|^2\right].$$

Instead of assuming strong convexity of the optimized loss $F_n$, we assume in the following that the Gibbs distribution with density proportional to $e^{-F_n}$ satisfies the LSI. A precise discussion on which conditions of $F_n$ yield the LSI is provided in section 5. Moreover, our main technical tool will rely on showing that the outputs $X_k$ of equation SGLD all satisfy the LSI with a constant that is uniform in $k$. It was unknown if such a uniform bound held without strong convexity. Vempala & Wibisono (2019) include a proof under strong-convexity in their last arxiv modification and Altschuler & Talwar (2022) specifically study this question under convexity. The lack of uniform LSI was the bottleneck that prevented analyses from capturing non-convex settings and required Vempala & Wibisono (2019) to state the uniform LSI as assumption(Assumption 2).

## 3 RELATED WORK

**Time-independent information-theoretic generalization**   Several authors have considered the question of time dependence of generalization bounds of noisy iterative schemes. The work of Mou et al. (2018) was the first to notice that each step includes a decay factor that can compensate for step-wise increases derived in previous analyses. Unfortunately, only a degrading decay factor is established, making vanishing step-sizes mandatory. Li et al. (2019) build on their result to show a time-independent bound for non-convex losses obtained as bounded perturbations of a strongly

convex loss. Unfortunately, their bounds do not go to 0 as $n \to \infty$. In bounded settings, where a projection step follows each noisy gradient update, the work of Wang et al. (2023) and Chien et al. (2024) establish that a uniform decay factor can be established. They show that if each iteration of SGLD is followed by a projection on a convex set $\mathcal{C}$, then a constant decay factor that depends on $\exp\left(\mathbf{diam}(\mathcal{C})/\eta\right)$ can be established.

Fewer papers tackle the unbounded setting. Using coupling techniques, Farghly & Rebeschini (2021) establish a time-uniform bound but incur inelegant step size dependences and do not obtain a bound going to 0 as $n \to \infty$ for fixed stepsizes. The recent work Zhu et al. (2024) considers the same dissipative setting and exploits Markov chain perturbation results but their Wassertein analysis requires that Lipschitz losses be used to measure the generalization gap. The work most closely related to ours is Futami & Fujisawa (2024). For dissipative losses, their result involves dimension-dependent quantities unrelated to stability. We show that the dissipative setting is friendly enough to not require such constants, and we improve their analysis to remove the dissipativity assumption under ergodicity. We include a table for ease of comparison in Appendix A.

## 4 ANALYSIS TEMPLATE

In this section, we describe the analysis template depicted in Figure 1 which will allow us to establish KL and Rényi stability through a step-wise analysis of each iteration $k$. Earlier versions of this template appear in Chourasia et al. (2021) and Ye & Shokri (2022)(appendix D.7), who refined the Rényi analysis of Vempala & Wibisono (2019). Our analysis relies on showing that, at each iteration, there is an expansion followed by a contraction.

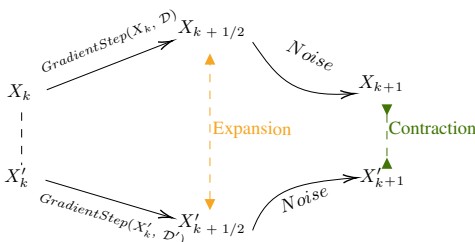

Figure 1: Analysis template

Before diving into the analysis, we ease the computations by assuming that the sequences of batches are chosen before the start of the recursion. In other words, we conduct the analysis conditioned on $B = (B_k)_k$ by using the fact that *conditioning increases $D_q$* (see 7.11 in Polyanskiy (2019)). That is, for any distributions $a, b$,

$$D_q\left(a\|b\right) \leq \mathbb{E}_B\left[D_q\left(a|B\|b|B\right)\right]$$

when $q \geq 1$. Consequently, we can conduct the analysis for a fixed non-random sequence of batches, and then take the expectation of the final result with respect to the batch selection. This is the standard simplification for analyzing stochastic gradients Wang et al. (2023); Ye & Shokri (2022).

The first step of our method consists of decomposing the noise term as follows

$$X_{k+1} = X_k - \eta g(X_k, B_k) + \sqrt{\frac{2\eta}{\beta}} N_{k+1}$$

$$= \underbrace{X_k - \eta g(X_k, B_k) + \sqrt{\frac{\eta}{\beta}} N_{k+1}^{(1)}}_{\text{Expansion term}} + \underbrace{\sqrt{\frac{\eta}{\beta}} N_{k+1}^{(2)}}_{\text{Contraction term}} \quad (3)$$

$$=: X_{k+1/2} + \sqrt{\frac{\eta}{\beta}} N_{k+1}^{(2)}$$

where we have split the random variable $N_{k+1}$ into two independent $\mathcal{N}(0, I)$ variables $N_{k+1}^{(1)}$ and $N_{k+1}^{(2)}$. A single update therefore corresponds to two consecutive half steps, the first going from $X_k$ to $X_{k+1/2}$ (the gradient update half-step) and a second going from $X_{k+1/2}$ to $X_{k+1}$ (the noise step). We analyze these half-steps in what follows for the iterates $(X_k)_k, (X_k')_k$ defined in equation 2.

### 4.1 EXPANSION HALF-STEP

The control of the divergence along the first half-step will result from the analog of the chain rule for $q$-Rényi divergence. We can show that the following bounded expansion holds.

**Theorem 5** (Bounded expansion). *Let $X_{k+1/2}$ and $X'_{k+1/2}$ be the gradient update half-steps in equation 3, then*

$$D_q\left(X_{k+1/2}\|X'_{k+1/2}\right) \leq D_q\left(X_k\|X'_k\right) + q\frac{\beta\eta}{2}\tilde{\mathbb{E}}_{k,q}[\|g(X'_k, B_k) - g'(\tilde{X}'_k, B'_k)\|_2^2].$$

*where the tilted expectation $\tilde{\mathbb{E}}_{k,q}$ is an expectation under a modified density defined in Definition 21.*

The result above tells us that the expansion (or over-fitting) induced by the gradient step can be controlled by the term

$$S_k := \tilde{\mathbb{E}}_{k,q}[\|g(X'_k, B_k) - g'(X'_k, B'_k)\|_2^2] \tag{4}$$

which is a quantity that measures how sensitive gradients are with respect to changes in the dataset.

## 4.2 CONTRACTION HALF-STEP

After the expansion half-step, the next iterates are obtained by simply adding the remaining half of the Gaussian noise. In other words, the next iterates are obtained after simultaneous Gaussian convolution (or diffusion along heat flow). This parallel addition of independent Gaussian noise (or Additive Gaussian noise channels) has been well explored in the sampling literature Wibisono & Jog (2018); Vempala & Wibisono (2019); Chewi et al. (2021). In particular Chen et al. (2022) generalize a result of Vempala & Wibisono (2019) to derive the following contraction theorem.

**Theorem 6** (Adapted from Chen et al. (2022) Theorem 3). *Let $X_{k+1/2}$ and $X'_{k+1/2}$ be the gradient half-steps as defined in equation 3. If $(X'_{k+1/2})_k$ all verify the LSI with contant $\alpha$, then after simultaneous heat flow $X_{k+1/2} + \sqrt{\eta/\beta}N$ and $X'_{k+1/2} + \sqrt{\eta/\beta}N'$, with $N, N' \sim \mathcal{N}(0, I)$, we have that*

$$D_q\left(X_{k+1}\|X'_{k+1}\right) \leq \gamma D_q\left(X_{k+1/2}\|X'_{k+1/2}\right).$$

*where $\gamma = \left(\frac{\beta\alpha}{\beta\alpha+\eta}\right)^{1/q} < 1$.*

If a uniform LSI can be established, the addition of noise after each gradient step corrects the over-fitting and brings the distributions back to being closer as shown in Figure 1. The proofs for this template can be found in B.

## 4.3 COMBINING THE STEPS

With the bounded expansion and approximate contraction results, it suffices to unroll the recurrence to obtain time-independent bounds. By combining theorems 5 and 6, we obtain the following single-step result.

**Theorem 7** (Single step bound). *Let $k \in \mathbb{N}$, $(X_k)$ and $(X'_k)$ the two sets of SGLD iterates defined equation 2. If $(X'_{k+1/2})_k$ all verify the LSI with contant $\alpha$, we have that*

$$D_q\left(X_{k+1}\|X'_{k+1}\right) \leq \gamma D_q\left(X_k\|X'_k\right) + \gamma q\frac{\beta\eta}{2}S_k$$

*where $S_k$ is the gradient sensitivity in equation 4 and $\gamma = \left(\frac{\beta\alpha}{\beta\alpha+\eta}\right)^{1/q} < 1$.*

Under bounded gradient sensitivity (equation 4), this geometric recurrence given above remains bounded for $k \to \infty$. Indeed a simple unrolling yields $D_q\left(X_k\|X'_k\right) \leq q\frac{\beta\eta}{2}\sum_{t=0}^{k}\gamma^{k-t}S_t$. The results thus all hinge on finding a constant $\alpha$ such that all iterates of equation SGLD verify the LSI inequality with constant $\alpha$.

## 5 UNIFORM LSI UNDER DISSIPATIVITY

In this section, we show that the iterates of equation SGLD all verify the LSI under a dissipativity assumption on $f$. For a target distribution $\pi \propto e^{-f}$, a set of increasingly relaxed structural assumptions can be made on $f$ to guarantee that $\pi$ admits an LSI constant. A hierarchy of the commonly

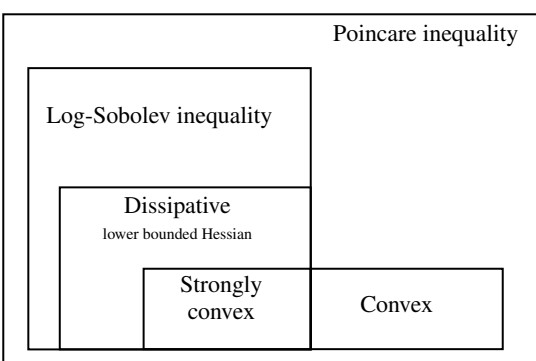

Figure 2: A diagram of the commonly used assumptions. A proof that strongly convex functions are dissipative can be derived from the quadratic lower bound at 0. Corollary 2.1.(2) of Cattiaux et al. (2010) shows that dissipativity and lower bounded Hessians imply LSI. The fact that Poincaré inequalities hold for log-concave measures is shown in Corollary 1.9 of Bakry et al. (2008). The proof that LSI implies Poincaré can be found in Bakry et al. (2014)(Proposition 5.1.3).

used assumptions is given in Figure 2. The iterates of equation SGLD however are evolving distributions so establishing that the structural assumptions hold uniformly for the distribution of the iterates $X_k$ can be burdensome. Luckily the log-Sobolev constant is stable through Lipschitz mappings and convolutions as shown below.

**Theorem 8** (Operations preserving LSI Chafaï (2004)). *If the distribution of a random variable $A$ is LSI($c_a$), then the distribution of $T(A)$ is LSI$\big(Lip(T)^2 c_a\big)$ and if $B$ is independent from $A$ and LSI($c_b$), then $A + B$ is LSI($c_a + c_b$).*

By leveraging these stability results, structural assumptions on the gradient mapping $T : x \mapsto x - \eta \nabla f(x)$ in SGLD can lead to uniform LSI constants. The following minimal assumption is necessary to show that the gradient mapping is Lipschitz.

**Assumption 9** (Smoothness). *For any $z \in \mathcal{Z}$, the function $x \mapsto f(x, z)$ is continuously twice differentiable and there exists $L > 0$ such that for any $z \in \mathcal{Z}$, $\|\nabla f(x, z) - \nabla f(y, z)\| \leq L\|x - y\|$.*

With the above, we can track the log-Sobolev constant throughout the iterations and additional assumptions are then added to ensure boundedness of the constant.

### 5.1 Strongly convex setting

In the literature, uniform LSI constants have only been established in strongly convex settings Ganesh & Talwar (2020); Ye & Shokri (2022); Chourasia et al. (2021); Vempala & Wibisono (2019). The uniform constants are obtained by noticing that if $f(\cdot, z)$ is $m$-strongly convex for all $z$, then $X_{k+1}$ is obtained by applying a $(1 - \eta m)$-Lipschitz gradient mapping to $X_k$ and adding independent Gaussian noise. Using the stability properties of the LSI constant, it can be shown that

$$c_{\text{LSI}}(X_{k+1}) \leq (1 - \eta m)^2 c_{\text{LSI}}(X_k) + \frac{\eta}{\beta}.$$

The geometric sequence stays bounded and yields the desired uniform bound on the LSI of $X_k$. Without strong convexity, the gradient cannot have a Lipschitz constant less than 1 for all $\eta \leq \frac{1}{L}$. Consequently, the geometric sequence derived by considering successive gradient updates and noise gives $c_{\text{LSI}}(X_{k+1}) \leq (1 + \eta L)^2 c_{\text{LS}}(X_k) + \frac{\eta}{\beta}$. which can only diverge exponentially as the iterate count increases. We will show that using different techniques, a finite bound can be established under a relaxation of strong convexity.

### 5.2 Dissipative functions

The standard way of relaxing strong convexity is by adding perturbations. A classic result establishes that bounded perturbations of strongly convex functions still satisfy the LSI, albeit with an exponential degradation Holley & Stroock (1988). More recent results show that other types of perturbations can preserve the LSI. For instance, if $V$ is strongly convex and $H$ a Lipschtiz function, then $e^{-V+H}$ still verifies the LSI Brigati & Pedrotti (2024). This setting corresponds precisely to Lipschitz losses with weight decay analyzed by Farghly & Rebeschini (2021).

Another seemingly different relaxation is the requirement that $f$ be strongly convex outside of a bounded region. In other words, the gradient of $f$ is strongly monotone outside of some ball. This

can be expressed by adding a slack $b > 0$ to the gradient monotonicity condition to yield

$$\langle x - y, \nabla f(x) - \nabla f(y) \rangle \geq m\|x - y\|^2 - b \tag{5}$$

for all pairs $x, y$. The above condition is known as strong-dissipativity Erdogdu et al. (2022). We note here that strongly dissipative functions are equivalent to bounded or Lipschitz perturbations of strongly convex functions (see Lem.1 of Ma et al. (2019) or Lem.2.4 of Brigati & Pedrotti (2024)).

A step further can be achieved by only requiring *one point strong convexity* and dropping the $y$ in equation equation 5 . This is akin to the relaxations of strong convexity analyzed in Karimi et al. (2016) (see their appendix A for a hierarchy). By relaxing equation 5, we obtain the set of dissipative functions defined as below.

**Definition 10** (Dissipativity). *A function $f$ is $(m, b)$ dissipative if for any $x \in \mathbb{R}^d$, we have that $\langle x, \nabla f(x) \rangle \geq m\|x\|^2 - b$.*

The condition appeared in Cattiaux et al. (2010) as a simple criterion to ensure existence of a finite LSI constant. It has then become the standard relaxation of strong-convexity in the sampling and noisy optimization literature Raginsky et al. (2017); Erdogdu et al. (2022). Indeed, when $F$ is dissipative, then $e^{-\beta F}$ can be shown to admit an LSI constant of the order $\mathcal{O}\left(\exp(\beta + d)\right)$.

Our result hinges on the fact that, despite being the broadest relaxation of strong convexity, dissipative functions still admit well-behaved gradient updates.

**Lemma 11** (Dissipative gradient updates are approximately contractive). *Let $f$ be an $L$-smooth, $(m, b)$-dissipative function, then for any $\eta \leq \frac{m}{2L^2}$,*

$$\|x - \eta \nabla f(x)\|^2 \leq \omega \|x\|^2 + \left(2\eta^2 L^2 R^2 + 2\eta b\right)$$

*with $\omega = (1 - \eta m)$ and $R = \sqrt{\frac{b}{m}}$.*

This approximate contraction property gives control of the norms of the iterates of equation SGLD. This allows to show that the gradient mapping under dissipativity ensures that the iterates remain sub-Gaussian. A remarkable result of Chen et al. (2021) then allows us to upgrade this sub-Gaussianity to a log-Sobolev inequality.

**Theorem 12** (Uniform LSI). *Let $X_0 \sim \mathcal{N}(0, \sqrt{\frac{\eta}{\beta}}I)$, and let $f$ be $(m, b)$-dissipative, $\frac{31}{32m} < \eta \leq \frac{m}{2L^2}$ the iterates of equation SGLD all verify a Poincare and log-Sobolev inequality with constants*

$$C_P \leq \frac{4\eta}{\beta} \exp\left(32\left(b + d + \eta\beta(LR)^2\right)\right) \quad \text{and} \quad C_{LSI} \leq 6C_P\left(32\left(b + d + \eta\beta(LR)^2\right)\right)$$

*where $R = \sqrt{b/m}$.*

The proof of this result can be found in appendix C. The bound on the log-Sobolev inequality of the iterates is exponential in dimension, but is of the same order as the LSI constant of the target distribution $e^{-\beta f}$ Raginsky et al. (2017). It is thus unlikely that the bound can be improved without additional assumptions. The constant factors in bounds on $\eta$ are loose and can be improved with clever uses of Young's inequality (see appendix D).

## 5.3 Corollaries under dissipativity

The results on dissipative functions allow us to derive the following immediate corollaries of Theorem 7. We first state our assumptions.

**Assumption 13** (Uniform dissipativity). *For all $z \in \mathcal{Z}$, the function $x \mapsto f(x, z)$ is $(m, b)$-dissipative and $L$-smooth.*

This ensures that the mini-batches are gradients of dissipative functions. The next assumption is a mild requirement needed to ensure that the sensitivity terms $S_k$ equation 4 can be controlled.

**Assumption 14** (Pseudo-Lipschitz). *There exists $\theta, D > 0$ such that for any $z, z' \in \mathcal{Z}$, $\|\nabla f(x, z) - \nabla f(x, z')\| \leq \theta\|x\| + D$*

**Corollary 14.1** (Bounded KL stability). *For $q = 1$, under Assumptions 13 and 14, for any $k \geq 1$ and $\frac{31}{32m} < \eta \leq \frac{m}{2L^2}$, we have that the iterates $X_k$ and $X'_k$ of SGLD with stay within a bounded KL divergence from each other given by*

$$D\left(X_k \| X'_k\right) \leq \frac{\beta\eta(\theta^2 M + D^2)}{(1 - \gamma)}\left(1 - \gamma^{k+1}\right)$$

*with $\gamma = \left(\frac{\beta\alpha}{\beta\alpha+\eta}\right)$, $M = \frac{2\eta L^2 R^2 + 2b}{m} + \frac{2d}{m\beta}$ and with $\alpha = (1 + \eta L)^2 C_{LSI} + \frac{\eta}{\beta}$, where $C_{LSI}$ is the uniform LSI given in Theorem 12.*

The pseudo-Lipschitz assumption which appears in Zhu et al. (2024) is merely an alternative way of bounding the sensitivity terms $S_k$ in equation 4, without requiring a uniform sensitivity bound when $q = 1$. For other divergences, the pseudo-Lipschitz assumption is insufficient, we need a more stringent $L_\infty$-bounded sensitivity assumption (which can be ensured by clipping gradients Ye & Shokri (2022)).

**Assumption 15** ($L_\infty$-bounded sensitivity). *For any $z, z' \in \mathcal{Z}$, $\|\nabla f(x, z) - \nabla f(x, z')\|^2 \leq S_\infty$*

**Corollary 15.1** (Rényi-differential privacy under dissipativity). *For $q \geq 1$, under Assumptions 13 and 15, for any $k \geq 1$ and $\frac{31}{32m} < \eta \leq \frac{m}{2L^2}$, we have that the iterates $X_k$ and $X'_k$ of SGLD stay within a bounded Rényi divergence from each other given by*

$$D_q\left(X_k \| X'_k\right) \leq q\frac{\beta\eta S_\infty}{2(1 - \gamma)}\left(1 - \gamma^{k+1}\right)$$

*with $\gamma = \left(\frac{\beta\alpha}{\beta\alpha+\eta}\right)^{1/q} < 1$ and $\alpha = (1 + \eta L)^2 C_{LSI} + \frac{\eta}{\beta}$, where $C_{LSI}$ is given in Theorem 12.*

Our corollaries above imply a time-independent bound for the expected generalization gap (Lemma 2) and the privacy loss (Lemma 3) under dissipativity. Our bounds solely involve stability-related constants, just like the strongly-convex bound of Chourasia et al. (2021)[Thm 3] and they decay to zero as $n \to \infty$. Fundamentally the real tool we used was the *upgrading* behavior of Gaussian convolution. In the next section, we show this is sufficient to remove dissipativity.

## 6 WITHOUT DISSIPATIVITY BUT WITH ERGODICITY

In this section, we show that dissipativity is not needed to establish time-uniform generalization bounds as long as the SGLD iterates converge towards a target distribution that verifies the LSI. It is possible to make milder requirements on $f$ at the cost of introducing quantities unrelated to stability: unlike Theorems 14.1 and 15.1, the result in this section will no longer match the strongly-convex lower-bound as it will include additional terms but its dimension dependence is improved. We significantly relax the analysis of Futami & Fujisawa (2024) which needlessly requires dissipativity and the parametrix method. We instead rely on simple tools to show that an approximate contraction result (Theorem 18) can replace Theorem 6. Instead of requiring a per-iterate LSI, Theorem 18 only requires the target to be LSI.

### 6.1 GAUSSIAN CONVOLUTION AND LOG-HESSIAN LOWER BOUNDS

The core of our result in this section relies on relaxing Theorem 6 in the analysis template. To do so, we must exchange distributions: we need to swap the per-iterate distribution with the target distribution. The swap is only possible if the distribution is sufficiently smooth. The following Lemma shows that Gaussian convolution enforces a lower bound on the Hessian of log-densities.

**Lemma 16** (log-Hessian lower bound). *Let $\nu$ be a distribution that results from a Gaussian convolution, i.e, $\nu = \tilde{\nu} \star \mathcal{N}(0, \eta I)$ for some distribution $\tilde{\nu}$, then $\nabla^2 \log \nu \succeq -\frac{1}{\eta}I_d$.*

This simple result follows from straightforward computations and what is more is that Gaussian convolution can only *improve* the log-Hessian lower bound of a distribution (see Lemma 35). Functions with a Hessian lower bound are convenient as they allow for simple changes of measure:

**Lemma 17** (Change of measure). *Let $g : \mathbb{R}^d \to \mathbb{R}$ be a twice-differentiable function such that $\nabla^2 g \succeq -K I_d$ for some $K \in \mathbb{R}$. Then, for any random variables $X, Y$ over $\mathbb{R}^d$, we have*

$$\mathbb{E}[-g(Y)] \leq \mathbb{E}[-g(X)] + \frac{1}{2}\mathbb{E}\left[\|\nabla g(X)\|\right] + \frac{K + 1}{2}\mathbb{E}[\|X - Y\|_2^2]$$

The lemma above gives us the ability to change an expectation under $Y$ to an expectation under $X$, and leaves open the choice of coupling between $X$ and $Y$ so we can make Wassertein distances appear on the left hand side by choosing optimal couplings. See Appendix F for the proofs.

## 6.2 Approximate contraction along simultaneous heat flow

We now have all the tools in hand to show that that if the SGLD chains converge to a well-behaved limit, then approximate contraction can be established. We begin by observing that the contraction step is applied to a half step $X_{k+1/2}$ that itself results from a Gaussian convolution since we split the noise in two. Consequently, we know that $\nabla^2 \log P_{X_{k+1/2}} \succeq \frac{\beta}{\eta} I_d$ by Lemma 34. The following theorem can then be established.

**Theorem 18** (Approximate Contraction)**.** *Let $\pi \sim e^{-\beta F_n}$ be a distribution verifying the LSI with constant $c_\pi \geq 1$, whose potential $F_n$ is $L$-smooth and lower bounded by $F^\star$. For any distribution $\pi'$ and $a_t = a_0 \star \mathcal{N}(0, t), b_t = b_0 \star \mathcal{N}(0, t)$ with $\nabla \log b_0 \succeq -\frac{\beta}{\eta} I$, we have that*

$$\mathrm{D}_{KL}\left(a_{\frac{\eta}{\beta}} \| b_{\frac{\eta}{\beta}}\right) \leq e^{-\eta/4c_\pi} \mathrm{D}_{KL}\left(a_0 \| b_0\right) + erg(a_\eta, b_\eta, \pi, \pi') + ProbConst$$

*where the ergodicity error term gathers quantities related to convergence of $a_\eta, b_\eta$ towards $\pi, \pi'$ and ProbConst gathers problem-dependent constants (explicitly given in equation 8 and equation 9).*

The approximate contraction established in Theorem 18, can be instantiated for our two parallel chains of SGLD to yield the following corollary under the following assumptions.

**Assumption 19** (Reasonable loss)**.** *For any dataset $\mathcal{D}$, the function $F_n$ is $L$-smooth and lower bounded by $F^\star \in \mathbb{R}$. The distribution $\pi \sim e^{-\beta F_n}$ verifies the LSI with constant $c_\pi$ and has bounded second moments $\mathbb{E}_\pi\left[\|X\|^2\right] < \infty$.*

**Assumption 20** (Bounded variance)**.** *The stochastic gradients are unbiased and satisfy $\mathbb{E}_B\left[\|g(X, B) - F_n(X)\|^2\right] \leq \sigma^2$.*

**Corollary 20.1** (KL stability under isoperimetry)**.** *Under assumption 19, 20, and assuming $c_\pi \geq 1$. We have for $\eta \leq \frac{\beta}{c_\pi L^2}$, the iterates of equation SGLD $X_k$ and $X_k'$ ran on datasets $\mathcal{D}$ and $\mathcal{D}'$ satisfy*

$$\mathrm{D}_{KL}\left(X_k \| X_k'\right) \leq \frac{poly\left(\frac{\eta}{\beta}, L, d, \sigma, \mathrm{D}_{KL}\left(X_0 \| \pi\right), \mathrm{D}_{KL}\left(X_0' \| \pi'\right)\right) + C_F + c_\pi^2 S_{Gibbs}}{1 - \gamma}\left(1 - \gamma^{k+1}\right)$$

*where $\gamma = e^{-\eta/4\beta c_\pi}$ and $S_{Gibbs} = \mathbb{E}_\pi\left[\|\nabla F_n(X) - \nabla F_n'(X)\|^2\right]$, and $C_F = \mathbb{E}_{\pi'}\left[\|X\|^2\right] - 2F^\star$.*

We prove this Corollary in Appendix E.1. We are able to show KL stability while making the fewest assumptions on the structure of the optimized loss $F_n$. Our result shows that merely knowing that $e^{-\beta F_n}$ verifies an isoperimetric inequality is sufficient to establish a generalization bound that does not degrade as the iteration count increases.

## 7 Conclusion

Our Rényi and KL stability bounds directly imply generalization and differential privacy guarantees for disspativite objectives, extending results only available for strongly convex settings. Noisy iterative algorithms can be run ad infinitum with non-vanishing step sizes without early-stopping in non-convex settings. This is in accordance with the practical observations that long training runs do not always harm generalization and in fact can sometimes improve it Olmin & Lindsten (2024). We also relax the dissipativity assumption to show a generalization bound that holds solely under an isoperimetric assumption of $F_n$.

The main limitation of our work is the dimension dependence of our bounds, which is affected by the choice of $\beta$. For the algorithm to be useful at minimization, a choice $\beta = \mathcal{O}(d)$ Raginsky et al. (2017) is necessary. However, the dependence of $c_\pi$ on $\beta$ is in general poor. Under additional assumption on $F_n$, this dependence can be made linear Li & Erdogdu (2023), but remains unavoidable for information-theoretic bounds Livni (2024). The extension of our analysis to non-isotropic, non-Gaussian, or state-dependent noise will inch closer to capturing SGD and further confirm that early-stopping is not a requirement in non-convex settings.

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

## A    RELATED WORK TABLE

Here we present and compare with some of the related work in table form for the reader's convenience.

| | non-convex | un-bounded domain | Bound |
|---|---|---|---|
| Chourasia et al. (2021); Ye & Shokri (2022) | ✗ | ✓ | $\mathcal{O}\left(1/\sqrt{n}\right)$ |
| Raginsky et al. (2017) | ✓ | ✓ | $\mathcal{O}\left(\eta K + \exp -\eta K/c_\pi + 1/n\right)$ |
| Mou et al. (2018) | ✓ | ✗ | $\mathcal{O}\left(\sqrt{\log K/n}\right)$ |
| Farghly & Rebeschini (2021) | ✓ | ✓ | $\mathcal{O}\left(1/(n\sqrt{\eta}) + \sqrt{\eta}\right)$ |
| Futami & Fujisawa (2024) | ✓ | ✓ | $\mathcal{O}\left(\sqrt{\frac{1+C}{n}}\right)$ with $C$ $d,b,m$-dependent |
| Wang et al. (2023); Chien et al. (2024) | ✓ | ✗ | $\begin{cases}\mathcal{O}\left(1/n\right) \text{ if } K \leq n \\ \mathcal{O}\left(\sqrt{1/n}\right) \text{ with (Chien et al., 2024)}\end{cases}$ |
| **Present work** | ✓ | ✓ | $\mathcal{O}\left(\sqrt{1/n}\right)$ |

Table 1: Our result matches the strongly-convex bound of Chourasia et al. (2021); Ye & Shokri (2022) in the **un-bounded, non-convex** setting.

Observe that the fast $1/n$ rate achievable with information theoretic bounds comes with drawbacks. Wang et al. (2023) achieve the fast rate **only for** $k \leq n$. In other words the result only holds for the first epoch of training. Other bounds obtaining the fast rate (see Rodríguez-Gálvez et al. (2024)) cannot be readily applied to SGLD to take into account the algorithm which is the primary goal of our work. It remains open to obtain a time-independent $1/n$ rate for SGLD (with no additional terms that do not decay to zero as in Farghly & Rebeschini (2021)).

## B    ANALYSIS TEMPLATE PROOFS

In this section we provide proofs for the results in the expansion-contraction template. The first result we prove is the bounded expansion result. We first define the tilted expectation.

**Definition 21** (q-tilted Expectation). *For any function h, the tilted expectation $\tilde{\mathbb{E}}_{k,q}$ is an expectation under a modified density and is defined by*

$$\tilde{\mathbb{E}}_{k,q}\left[h(X'_k)\right] = \mathbb{E}_{X'_k}\left[\phi_q(X'_k)h(X'_k)\right]$$

*with tilting function is the ratio $\phi_q := \frac{1}{\Lambda_q}\left(\frac{p_{X_k}}{p_{X'_k}}\right)^q$ where the normalization is given by $\Lambda_q = \mathbb{E}_{X'_k}\left[\left(\frac{p_{X_k}}{p_{X'_k}}\right)^q\right]$. If $q = 1$, the tilted expectation simplifies to*

$$\tilde{\mathbb{E}}_{k,q}\left[h(X'_k)\right] = \mathbb{E}_{X_k}\left[h(X_k)\right].$$

We can show that the over-fitting induced at each step is controlled by this tilted expectation as shown in the following theorem.

**Theorem 5** (Bounded expansion). *Let $X_{k+1/2}$ and $X'_{k+1/2}$ be the gradient update half-steps in equation 3, then*

$$D_q\left(X_{k+1/2}\|X'_{k+1/2}\right) \leq D_q\left(X_k\|X'_k\right) + q\frac{\beta\eta}{2}\tilde{\mathbb{E}}_{k,q}[\|g(X'_k, B_k) - g'(\tilde{X}'_k, B'_k)\|_2^2].$$

*where the tilted expectation $\tilde{\mathbb{E}}_{k,q}$ is an expectation under a modified density defined in Definition 21.*

*Proof.* We first apply the data processing inequality to obtain that

$$D_q\left(X_{k+1/2}\|X'_{k+1/2}\right) \leq D_q\left((X_{k+1/2}, X_k)\|(X'_{k+1/2}, X'_k)\right)$$

The $q$-tilted chain rule for $q$-Rényi divergences ( (7.59) in Polyanskiy (2019)) then gives us

$$D_q\left((X_{k+1/2}, X_k)\|(X'_{k+1/2}, X'_k)\right) \leq D_q\left(X_k\|X'_k\right) + \tilde{\mathbb{E}}_{\tilde{X}_k}D_q\left(X_{k+1/2}|X_k = \tilde{X}_k\|X'_{k+1/2}|X'_k = \tilde{X}_k\right)$$

where the tilted expectation is given by

$$\tilde{\mathbb{E}}_{\tilde{X}'_k} \left[ h(\tilde{X}'_k) \right] = \mathbb{E}_{X'_k} \left[ \phi_q(X'_k) h(X'_k) \right]$$

where $\phi_q(x) = \frac{1}{\Lambda_q} \left( \frac{p_{X_k}}{p_{X'_k}} \right)^q p_{X'_k}$ where the normalization constant is given by $\Lambda_q = \mathbb{E}_{X'k} \left[ \left( \frac{p_{X_k}}{p_{X'_k}} \right)^q \right]$. This tilted expectation is a little complicated but what is inside the expectation is a simple term.

Observe that $X_{k+1/2}|X_k = x$ is a $\mathcal{N}(x - g(x, B_k), \sqrt{\frac{\eta}{\beta}})$. We obtained the result by using closed form results for the $q$-Rényi divergence between Gaussians (Mironov (2017) Proposition 7). $\qquad\square$

The second element is the contraction component. The contraction theorem 6 is a direct application of the forward step in Theorem 3 of Chen et al. (2022) with $t = \eta/\beta$.

## C    Uniform LSI under dissipativity

In this section we prove that the iterates of equation SGLD verify a uniform log-Sobolev inequality under assumption 10.

### C.1    Properties of dissipative functions

We first begin by showing approximative contraction property of dissipative functions.

**Lemma 22** (Dissipative gradient updates are approximately contractive). *Let $f$ be an $L$-smooth, $(m, b)$-dissipative function, then for any $\eta \leq \frac{m}{2L^2}$,*

$$\|x - \eta \nabla f(x)\|^2 \leq \omega \|x\|^2 + \left( 2\eta^2 L^2 R^2 + 2\eta b \right)$$

*with $\omega = (1 - \eta m)$ and $R = \sqrt{\frac{b}{m}}$.*

*Proof.* Let $\eta \leq \frac{\sqrt{m}}{2L}$, let $x \in \mathbb{R}^d$,

$$\|x - \eta \nabla f(x)\|^2 = \|x\|^2 - 2\eta \langle x, \nabla f(x) \rangle + \eta^2 \|\nabla f(x)\|^2$$
$$\leq \|x\|^2 - 2\eta m \|x\|^2 + 2\eta^2 L^2 \|x\|^2 + 2\eta^2 L^2 \frac{b}{m} + 2\eta b \quad \text{(using Ass. 10 and Lemma 23)}$$
$$\leq (1 - \eta m) \|x\|^2 + 2\eta^2 L^2 \frac{b}{m} + 2\eta b \quad \text{(using that } \eta \leq \frac{m}{2L^2} \text{)}$$
$$\leq \omega \|x\|^2 + 2\eta^2 L^2 R^2 + 2\eta b$$

with $\omega = (1 - \eta m)$ and $R = \sqrt{\frac{b}{m}}$. $\qquad\square$

**Lemma 23** (Gradients of dissipative and smooth functions). *Let $f$ be an $L$-smooth, $(m, b)$-dissipative function, then*

$$\|\nabla f(x)\| \leq L\|x\| + L\sqrt{\frac{b}{m}}$$

*Proof.* Let $x^\star$ be a stationary point of $f$, then

$$\|\nabla f(x)\| = \|\nabla f(x) - \nabla f(x^\star)\| \leq L\|x - x^\star\| \leq L\|x\| + L\|x^\star\|$$

The result follows from Lemma 24. $\qquad\square$

**Lemma 24** (Stationary points of dissipative functions). *Let $f$ be an $(m, b)$-dissipative function, then for any $x \in \mathbb{R}^d$,*

$$\nabla f(x) = 0 \implies \|x\|^2 \leq \frac{b}{m}$$

*Proof.* The result follows from the definition of dissipativity in 10. $\qquad\square$

Dissipative functions therefore roughly keep the iterates in a bounded region of size $R = \sqrt{\frac{b}{m}}$. This will have implications for their exponential integrability.

## C.2 EXPONENTIAL INTEGRABILITY

Given that each gradient update is contracting, we can show that the iterates of SGLD are sub-Gaussian. In other words, we can show exponential integrability as given by the following lemma.

**Lemma 25** (Exponential integrability). *Let $X_0 \sim \mathcal{N}(0, \sqrt{\frac{\eta}{d\beta}}I)$, and let $f$ be $(m, b)$-dissipative with contraction constant $\omega = (1 - \eta m)$. For any $p \geq 1$ such that $p\omega < 1/8$, we have for any $k \geq 1$, the iterate $X_k$ of equation SGLD verifies*

$$\frac{1}{p} \log \mathbb{E} \left[ \exp \left( 2p \frac{\beta}{2\eta} \|X_k - \eta \nabla f(X_k)\|^2 \right) \right] \leq 16 \left( d + \eta \beta (LR)^2 \right) p$$

*Proof.* We show the result by expressing the norm of the gradient update as a sum of Gaussian norms. Let $k \geq 1$, we have that

$$\|X_k - \eta \nabla f(X_k)\|^2 \leq \omega \|X_k\|^2 + 2\eta^2 L^2 R^2 + 2\eta b$$

$$\leq 2\omega \|X_{k-1} - \eta \nabla f(X_{k-1})\|^2 + 2\omega \frac{2\eta}{\beta} \|N_k\|^2 + 2\eta^2 L^2 R^2 + 2\eta b$$

$$\leq (2\omega)^k \|X_0\|^2 + \frac{2\eta}{\beta} \sum_{i=0}^{k} (2\omega)^{i+1} \|N_{k-i}\|^2 + \frac{2\eta^2 L^2 R^2 + 2\eta b}{1 - 2\omega}$$

Since $X_0 \sim \mathcal{N}(0, \sqrt{\frac{\eta}{d\beta}}I)$, we have that

$$\frac{\beta}{2\eta} \|X_k - \eta \nabla f(X_k)\|^2 \leq \frac{1}{d} (2\omega)^k \|N_0\|^2 + \sum_{i=0}^{k} (2\omega)^{i+1} \|N_{k-i}\|^2 + \frac{\eta \beta (LR)^2 + b}{2\eta m - 1}$$

where each $N_i$ is an independent $\mathcal{N}(0, I)$ variable. Norms of Gaussians are exponentially integrable. Consequently, for $p \geq 1$, such that $2p(2\omega) < 1/4$, we can invoke Lemma 26 to find that

$$\log \mathbb{E} \left[ \exp \left( 2p \frac{\beta}{2\eta} \|X_k - \eta \nabla f(X_k)\|^2 \right) \right] \leq \left( 4p(2\omega)^k + 4 \sum_{i=0}^{k} (2\omega)^i dp + \frac{\eta \beta (LR)^2 + b}{2\eta m - 1} \right)$$

$$\leq 16 \left( b + d + \eta \beta (LR)^2 \right) p$$

where we used the fact that $\omega < 1/4$ implies that $2\eta m - 1 \geq 1/2$. $\qquad\square$

**Lemma 26** (Square Gaussian moment generating function). *Let $Z \sim \mathcal{N}(0, I_d)$, we have that, for any $\lambda < 1/4$,*

$$\log \mathbb{E} \left[ e^{\lambda \|Z\|^2} \right] \leq 2d\lambda$$

*Proof.* The random variable $\|Z\|^2$ is a sum of $d$ standard Gaussians squared. Using elementary computations we can derive its moment generating function

$$\mathbb{E}[e^{\lambda \|Z\|^2}] = \left( \frac{1}{1 - 2\lambda} \right)^{d/2}$$

We obtain the result using the inequality $-\ln(1 - 2\lambda) \leq 4\lambda$ for $\lambda < 1/4$. $\qquad\square$

With the Exponential integrability, we simply exploit the fact that Gaussian convolution upgrades sub-Gaussianity to an LSI to show the following result.

**Theorem 12** (Uniform LSI). *Let $X_0 \sim \mathcal{N}(0, \sqrt{\frac{\eta}{\beta}}I)$, and let $f$ be $(m,b)$-dissipative, $\frac{31}{32m} < \eta \leq \frac{m}{2L^2}$ the iterates of equation SGLD all verify a Poincare and log-Sobolev inequality with constants*

$$C_P \leq \frac{4\eta}{\beta} \exp\left(32\left(b + d + \eta\beta(LR)^2\right)\right) \quad and \quad C_{LSI} \leq 6C_P\left(32\left(b + d + \eta\beta(LR)^2\right)\right)$$

*where $R = \sqrt{b/m}$.*

*Proof.* Thanks to our Lemma 25, the proof will follow from the fact that Gaussian convolution upgrades sub-Gaussianity to a log-Sobolev inequality which was shown in Chen et al. (2021). For any $k \geq 1$, we have that

$$X_k = \underbrace{X_k - \eta g(X_k, B_k)}_{\text{Exponentially integral gradient step}} + \underbrace{\sqrt{\eta}N_{k+1}}_{\text{Independent noise}}$$

We cast our result in their notation. We can express the equation above as the mixture of Gaussians. Indeed if $\mu$ denotes the distribution of the gradient step and $P_x$ is the Gaussian distribution centered at $X$, then the distribution of $X_k$ is the mixture $\mu P := \int P_x d\mu(x)$. We can thus apply Theorem 1 Chen et al. (2021) which bounds the LSI of mixtures. Using the exponential integrability bound established in Lemma 25 and taking $p = 2$, we have, following their notation, that,

$$K_{p,\chi^2}(P,\mu) := \mathbb{E}_{X\sim\mu, X\sim\mu'}\left[1 + \chi^2\left(P_X \| P_X'\right)\right] \leq \mathbb{E}\left[\exp\left(4\frac{\beta}{2\eta}\|X_k - \eta\nabla f(X_k)\|^2\right)\right]$$

By combining our bound in Lemma 25 and Theorem 1 Chen et al. (2021) yields the result since $K_P = \frac{\eta}{\beta}$. □

## C.3 COROLLARIES UNDER DISSIPATIVITY

Here we provide proofs for the corollaries derived under dissipativity.

**Corollary 14.1** (Bounded KL stability). *For $q = 1$, under Assumptions 13 and 14, for any $k \geq 1$ and $\frac{31}{32m} < \eta \leq \frac{m}{2L^2}$, we have that the iterates $X_k$ and $X_k'$ of SGLD with stay within a bounded KL divergence from each other given by*

$$D\left(X_k \| X_k'\right) \leq \frac{\beta\eta(\theta^2 M + D^2)}{(1-\gamma)}\left(1 - \gamma^{k+1}\right)$$

*with $\gamma = \left(\frac{\beta\alpha}{\beta\alpha+\eta}\right)$, $M = \frac{2\eta L^2 R^2 + 2b}{m} + \frac{2d}{m\beta}$ and with $\alpha = (1+\eta L)^2 C_{LSI} + \frac{\eta}{\beta}$, where $C_{LSI}$ is the uniform LSI given in Theorem 12.*

*Proof.* To establish the corollary, it suffices to control the sensitivity terms $S_k$ given in equation 4. Thanks to assumption 14, we have that

$$S_k \leq 2\theta^2 \tilde{E}\left[\|X_k'\|^2\right] + 2D^2$$

Since we have chosen $q = 1$, the tilted expectation is actually an expectation under $X_k$ according to Definition 21. Consequently,

$$\tilde{E}\left[\|X_k'\|^2\right] = E\left[\|X_k\|^2\right] \leq \omega E\left[\|X_{k-1}\|^2\right] + \left(2\eta^2 L^2 R^2 + 2\eta b\right) + \frac{2d\eta}{\beta}$$

where the second inequality follows from the approximate contractions of dissipative gradient updates Lemma 11 and the last term is the expected norm of Gaussian noise $N_k$. By unrolling the geometric sequence above, we have that

$$E\left[\|X_k\|^2\right] \leq \frac{1}{1-\omega}\left[\left(2\eta^2 L^2 R^2 + 2\eta b\right) + \frac{2d\eta}{\beta}\right]$$

Using the fact that $1 - \omega = \eta m$, we find that

$$E\left[\|X_k\|^2\right] \leq \left(\frac{2\eta L^2 R^2 + 2b}{m} + \frac{2d}{m\beta}\right)$$

Defining $M := \frac{2\eta L^2 R^2 + 2b}{m} + \frac{2d}{m\beta}$ yields the result. □

## D    OPTIMIZING CONSTANTS

With regards to optimizing the constant $31/32$, we first observe that for any $x \in \mathbb{R}^d$, and $\iota > 0$, we have thanks to Lemma 23,

$$\eta^2 |\nabla f(x)|^2 \leq \eta^2 \left( L|x| + L\sqrt{\frac{b}{m}} \right)^2$$

$$= \eta^2 \left( L^2 |x|_2^2 + 2(L|x|)(L\sqrt{\frac{b}{m}}) + L^2 \frac{b}{m} \right)$$

$$= \eta^2 \left( L^2 |x|_2^2 + 2(\sqrt{\iota} L|x|)(\frac{1}{\sqrt{\iota}} L\sqrt{\frac{b}{m}}) + L^2 \frac{b}{m} \right)$$

$$\leq (1+\iota)\eta^2 L^2 |x|_2^2 + (1+\frac{1}{\iota})\eta^2 L^2 \frac{b}{m}$$

where we have used Young's inequality in the last inequality. Now here, notice that we have $(1 + \iota)$ instead of 2 in the proof of Lemma 11. So we can choose $\eta \leq \frac{m}{(1+\iota)L^2}$, to have that

$$\eta^2 |\nabla f(x)|^2 \leq \eta m |x|_2^2 + (1+\frac{1}{\iota})\eta^2 L^2 \frac{b}{m}$$

By having a larger constant term, we can allow ourselves a larger choice of $\eta$. We can combine this result with dissipativity to obtain the following gradient contraction

$$|x - \eta \nabla f(x)|_2^2 \leq (1 - \eta m)|x|^2 + 2\eta^2 L^2 (1 + \frac{1}{\iota})\frac{b}{m} + 2\eta b$$

which tightened version of Lemma 11.

## E    APPROXIMATE CONTRACTION PROOFS

In this section, we prove the approximate contraction along heat flow theorem established in section.

**Theorem 18** (Approximate Contraction). *Let $\pi \sim e^{-\beta F_n}$ be a distribution verifying the LSI with constant $c_\pi \geq 1$, whose potential $F_n$ is L-smooth and lower bounded by $F^\star$. For any distribution $\pi'$ and $a_t = a_0 \star \mathcal{N}(0,t), b_t = b_0 \star \mathcal{N}(0,t)$ with $\nabla \log b_0 \succeq -\frac{\beta}{\eta}I$, we have that*

$$\mathrm{D}_{KL}\left(a_{\frac{\eta}{\beta}} \| b_{\frac{\eta}{\beta}}\right) \leq e^{-\eta/4c_\pi} \mathrm{D}_{KL}\left(a_0 \| b_0\right) + erg(a_\eta, b_\eta, \pi, \pi') + ProbConst$$

*where the ergodicity error term gathers quantities related to convergence of $a_\eta, b_\eta$ towards $\pi, \pi'$ and ProbConst gathers problem-dependent constants (explicitly given in equation 8 and equation 9).*

*Proof.* From Lemma 39, we know that

$$\frac{d}{dt}\mathrm{D}_{\mathrm{KL}}\left(a_t \| b_t\right) \leq -\frac{1}{2}\mathbb{E}\left[\|\nabla \log a_t(A_t) - \nabla \log b_t(A_t)\|^2\right]$$

Expanding the square, we find that

$$\frac{d}{dt}\mathrm{D}_{\mathrm{KL}}\left(A_t \| B_t\right) \leq -\frac{1}{2}\mathbb{E}_{a_t}\left[\|\nabla \log a_t\|^2\right] - \frac{1}{2}\mathbb{E}\left[\|\nabla \log b_t(A_t)\|^2\right] + \mathbb{E}_{a_t}\left[\langle \nabla \log a_t, \nabla \log b_t \rangle\right]$$

Now making the same observation as Futami & Fujisawa (2024) that $-\|u-v\|^2 \leq -\frac{1}{2}\|u\|^2 + \|v\|^2$, we have that

$$\frac{d}{dt}\mathrm{D}_{\mathrm{KL}}\left(A_t \| B_t\right) \leq -\frac{1}{4}\mathbb{E}_{a_t}\left[\|\nabla \log a_t - \nabla \log \pi\|^2\right] + \frac{1}{2}\mathbb{E}_{a_t}\left[\|\nabla \log \pi\|\right] - \frac{1}{2}\mathbb{E}_{a_t}\left[\|\nabla \log b_t\|^2\right] + \mathbb{E}_{a_t}\left[\langle \nabla \log a_t, \nabla \log b_t \rangle\right]$$

Invoking the LSI for $\pi$, we can write

$$\frac{d}{dt}\mathrm{D}_{\mathrm{KL}}\left(A_t \| B_t\right) \leq -\frac{1}{4c_\pi}\mathbb{E}_{a_t}\left[\log \frac{a_t}{\pi}\right] + \frac{1}{2}\mathbb{E}_{a_t}\left[\|\nabla \log \pi\|^2\right] - \frac{1}{2}\mathbb{E}_{a_t}\left[\|\nabla \log b_t\|^2\right] + \mathbb{E}_{a_t}\left[\langle \nabla \log a_t, \nabla \log b_t \rangle\right]$$

Introducing $b_t$ back into the first term, we find that

$$\frac{d}{dt} D_{KL}(A_t \| B_t) \leq -\frac{1}{4c_\pi} \mathbb{E}_{a_t} \left[ \log \frac{a_t}{b_t} \right] + \mathrm{Err}(\pi) + \mathrm{Err}(b) \tag{6}$$

where we define

$$\mathrm{Err}(\pi) := \frac{1}{4c_\pi} \mathbb{E}_{a_t}[\log \pi] + \frac{1}{2} \mathbb{E}_{a_t} \left[ \|\nabla \log \pi\|^2 \right]$$

and

$$\mathrm{Err}(b) := \frac{1}{4c_\pi} \mathbb{E}_{a_t}[-\log b_t] - \frac{1}{2} \mathbb{E}_{a_t} \left[ \|\nabla \log b_t\|^2 \right] + \mathbb{E}_{a_t} [\langle \nabla \log a_t, \nabla \log b_t \rangle]$$

We will establish time-independent upper bounds for both these error terms. Using Grónwall's lemma, we can then deduce from equation 6, by integrating from 0 to $t = \frac{\eta}{\beta}$ that

$$D_{KL}\left( a_{\frac{\eta}{\beta}} \| b_{\frac{\eta}{\beta}} \right) \leq e^{-\eta/4\beta c_\pi} D_{KL}(a_0 \| b_0) + \frac{\eta}{\beta} (\mathrm{Err}(\pi) + \mathrm{Err}(b)) \tag{7}$$

We control each term individually. First, observe that $\mathrm{Err}(\pi)$ is a sum of a negative Shannon entropy and a log-gradient norm measured through a distribution $a_t$ instead of $\pi$. We bound each summand in Lemmas 30 and 31 respectively to obtain that

$$\mathrm{Err}(\pi) \leq -2\beta F^\star + \frac{d}{2} \log(\frac{\beta L}{2\pi}) + 2\beta^2 L^2 \mathcal{W}_2^2 \left( a_{\frac{\eta}{\beta}}, \pi \right) + 2\beta^2 L^2 \eta + Ld$$

The other error term is where the regularizing properties of Gaussian convolution are fully exploited and it is *here that we differ most sharply with the analysis of Futami & Fujisawa (2024)*. The first term in $\mathrm{Err}(b)$ is an expectation of $-\log b_t$ and we have established that thanks to Gaussian convolution, the log-Hessian of $\log b_t$ is lower bounded. This induced smoothness allows us to perform a change of measure in Lemma 28 to control the first two terms of $\mathrm{Err}(b)$. The last term is again controlled using the properties of Gaussian convolution and simple integration by parts in Lemma 27. We obtain that

$$\mathrm{Err}(b) \leq \max_{t \leq \frac{\eta}{\beta}} \mathbb{E}_{b_t}[-\log b_t] + \frac{K+1}{2} \left( \mathcal{W}_2^2 \left( a_{\frac{\eta}{\beta}}, b_{\frac{\eta}{\beta}} \right) + 8\frac{\eta}{\beta}d \right) + Kd$$

Using the density bound in equation 29, we can control the Shannon entropy, which gives for any $\pi'$,

$$\mathrm{Err}(b) \leq \frac{4\beta}{\eta} \mathcal{W}_2^2 \left( b_{\frac{\eta}{\beta}}, \pi' \right) + \frac{4\beta}{\eta} \mathbb{E}_{\pi'} \left[ \|X\|^2 \right] + 2d + \frac{K+1}{2} \left( \mathcal{W}_2^2 \left( a_{\frac{\eta}{\beta}}, b_{\frac{\eta}{\beta}} \right) + 8\frac{\eta}{\beta}d \right) + Kd$$

Putting everything together in equation 7, with $K = \frac{\beta}{\eta}$, we find that

$$D_{KL}\left( a_{\frac{\eta}{\beta}} \| b_{\frac{\eta}{\beta}} \right) \leq e^{-\eta/4c_\pi} D_{KL}(a_0 \| b_0) + \mathrm{erg}(a_{\frac{\eta}{\beta}}, b_{\frac{\eta}{\beta}}, \pi, \pi') + \mathrm{ProbConst}$$

where the ergodicity error term gathers quantities related to the convergence of the processes

$$\mathrm{erg}(a_{\frac{\eta}{\beta}}, b_{\frac{\eta}{\beta}}, \pi, \pi') = 2\eta\beta^2 L^2 \mathcal{W}_2^2 \left( a_{\frac{\eta}{\beta}}, \pi \right) + (1 + \frac{\eta}{2\beta}) \mathcal{W}_2^2 \left( a_{\frac{\eta}{\beta}}, b_{\frac{\eta}{\beta}} \right) + 4\mathcal{W}_2^2 \left( b_{\frac{\eta}{\beta}}, \pi' \right) \tag{8}$$

and the problem constants intervene in ProbConst with

$$\mathrm{ProbConst} = \mathbb{E}_{\pi'} \left[ \|X\|^2 \right] - 2F^\star + \frac{\eta d}{2} \log(\frac{\beta L}{2\pi}) + \frac{d}{2} \log(2\pi\frac{\eta}{\beta}) + d \left( \eta^2 L^2 + Ld + d + 2\frac{\eta}{\beta} \right) \tag{9}$$

$\square$

**Lemma 27** (Inner product bound). *Let $a_t$, $b_t$ as in Theorem 18 with $\nabla^2 \log b_0 \succeq -K$,*

$$\mathbb{E}_{a_t} [\langle \nabla \log a_t, \nabla \log b_t \rangle] = -\mathbb{E}_{a_t} [\Delta \log b_t] \leq Kd$$

*Proof.* The result follows from integration by parts (Lemma 40) which gives

$$\mathbb{E}_{a_t} [\langle \nabla \log a_t, \nabla \log b_t \rangle] = -\mathbb{E}_{a_t} [\Delta \log b_t]$$

Since Gaussian convolution only improves log-Hessian lower bounds, we find that

$$-\mathbb{E}_{a_t} [\Delta \log b_t] \leq Kd$$

$\square$

**Lemma 28** (Change of measure). *Let $a_t$, $b_t$ as in Theorem 18, then for $c_\pi \geq 1$*

$$\frac{1}{4c_\pi}\mathbb{E}_{a_t}\left[-\log b_t\right] - \frac{1}{2}\mathbb{E}_{a_t}\left[\|\nabla \log b_t\|^2\right] \leq \mathbb{E}_{b_t}\left[-\log b_t\right] + \frac{K+1}{2}\left(\mathcal{W}_2^2\left(A_{\frac{\eta}{\beta}}, B_{\frac{\eta}{\beta}}\right) + 8\frac{\eta}{\beta}d\right)$$

*Proof.* Since $-\nabla^2 \log b_t \preceq K$, we can apply the change of measure Lemma 17 to the function $-\log b_t$ to find that for $c_\pi \geq 1$, we have

$$\frac{1}{4c_\pi}\mathbb{E}_{a_t}\left[-\log b_t\right] - \frac{1}{2}\mathbb{E}_{a_t}\left[\|\nabla \log b_t\|^2\right] \leq \mathbb{E}_{b_t}\left[-\log b_t\right] + \frac{K+1}{8}\mathbb{E}\left[\|A_t - B_t\|^2\right]$$

The crucial feature of the change of measure lemma is that we can choose the coupling between $(A_t, B_t)$ freely. Let $((A_0, N), ((B_0, N'))$ be coupled such that

$$\mathcal{W}_2^2\left(A_0 + \sqrt{\frac{\eta}{\beta}}N, B_0 + \sqrt{\frac{\eta}{\beta}}N'\right) = \mathbb{E}\left[\|A_0 + \sqrt{\eta}N - (B_0 + \sqrt{\eta}N')\|^2\right]$$

With this coupling in hand, we define $A_t = A_0 + \sqrt{t}Z$ and $B_t = B_0 + \sqrt{t}Z'$ with $Z, Z'$ independent $\mathcal{N}(0, I)$ variables. We then compute

$$\mathbb{E}\left[\|A_t - B_t\|^2\right] \leq 2\mathbb{E}\left[\|A_0 + \sqrt{\frac{\eta}{\beta}}N - (B_0 + \sqrt{\frac{\eta}{\beta}}N')\|^2\right] + 4td + 4\frac{\eta}{\beta}d$$

$$= \mathcal{W}_2^2\left(A_{\frac{\eta}{\beta}}, B_{\frac{\eta}{\beta}}\right) + 4(t + \frac{\eta}{\beta})d$$

$\square$

**Lemma 29** (Density bound). *Let $t \leq \eta \leq 1$, let $B$ be a random variable with density $b$. Let us define the half step density[2] as $b_{1/2} = (b \star \mathcal{N}(0, \eta/\beta I))$. Let $b_t = b_{1/2} \star \mathcal{N}(0, tI)$ be the result of a Gaussian convolution applied to the half step $b_{1/2}$, then for any $\pi'$, we have that*

$$\mathbb{E}_{b_t}\left[-\log b_t\right] \leq \frac{4\beta}{\eta}\mathcal{W}_2^2\left(b_{\frac{\eta}{\beta}}, \pi'\right) + \frac{4\beta}{\eta}\mathbb{E}_{\pi'}\left[\|X\|^2\right] + \frac{d}{2}\log(2\pi\frac{\eta}{\beta}) + 2d$$

*Proof.* By definition of Gaussian convolution, the following equalities hold

$$b_t(x) = \mathbb{E}_N\left[b_{1/2}(x - \sqrt{t}N)\right] \quad \text{and} \quad b_{1/2}(x) = \mathbb{E}_b\left[\mathcal{N}(x - B, \eta I)\right]$$

A repeated application of Jensen's inequality yields

$$-\log b_t(x) \leq \mathbb{E}_N\left[-\log b_{1/2}(x - \sqrt{t}N)\right]$$

$$\leq \frac{\beta}{2\eta}\mathbb{E}_N\mathbb{E}_b\left[\|x - \sqrt{t}N - B\|^2\right] + \frac{d}{2}\log(2\pi\frac{\eta}{\beta})$$

$$\leq \frac{\beta}{2\eta}\mathbb{E}_N\mathbb{E}_b\left[\|x - \sqrt{t}N - B\|^2\right] + \frac{d}{2}\log(2\pi\frac{\eta}{\beta})$$

Now taking expectation with respect to $b_t$, we find that

$$\mathbb{E}_{b_t}\left[-\log b_t(B_t)\right] \leq \frac{\beta}{2\eta}\mathbb{E}\left[\|B_t - \sqrt{t}N - B\|^2\right] + \frac{d}{2}\log(2\pi\frac{\eta}{\beta}) \tag{10}$$

We then add some noise terms to look forward in the heat flow. Let $N', N'' \sim \mathcal{N}(0, I)$ be independent Gaussians, then

$$\mathbb{E}\left[\|B_t - \sqrt{t}N - B\|^2\right] \leq \mathbb{E}\left[\|(B_t + \sqrt{\frac{\eta}{\beta} - t}N') - (B + \sqrt{2\eta}N'')\|^2\right] + (\frac{\eta}{\beta} - t)d + td + 2\eta d$$

$$\leq 2\mathbb{E}\left[\|B_{\frac{\eta}{\beta}}\|^2\right] + 2\mathbb{E}\left[\|B + \sqrt{2\eta}N''\|^2\right] + 3\frac{\eta}{\beta}d$$

---

[2](i.e akin to the density of $X_{k+1/2}$)

Recall that both $B_{\frac{\eta}{\beta}}$ and $B + \sqrt{2\frac{\eta}{\beta}}N''$ have the same law. Indeed $B_{\frac{\eta}{\beta}}$ has distribution $b_{\frac{\eta}{\beta}} = b_{1/2} \star \mathcal{N}(0, \frac{\eta}{\beta}I)$ (i.e $b_t$ with $t = \frac{\eta}{\beta}$) which adds variance $\frac{\eta}{\beta}$ gaussian noise to the half step which already adds variance $\frac{\eta}{\beta}$ gaussian noise to $B$, consequently it corresponds to adding $2\frac{\eta}{\beta}$ variance gaussian noise to $B$. By plugging the above into 10 that,

$$\mathbb{E}_{b_t}\left[-\log b_t(B_t)\right] \leq \frac{2\beta}{\eta}\mathbb{E}\left[\|B_{\frac{\eta}{\beta}}\|^2\right] + \frac{d}{2}\log(2\pi\frac{\eta}{\beta}) + \frac{3}{2}d$$

From this we deduce that,

$$\mathbb{E}_{b_t}\left[-\log b_t(B_t)\right] \leq \frac{4\beta}{\eta}\mathcal{W}_2^2\left(b_{\frac{\eta}{\beta}}, \pi'\right) + \frac{4\beta}{\eta}\mathbb{E}_{\pi'}\left[\|X\|^2\right] + \frac{d}{2}\log(2\pi\frac{\eta}{\beta}) + 2d.$$

$\square$

**Lemma 30** (Entropy bound lower bound). *Let $\pi \propto e^{-\beta F}$ with $F$ an $L$-smooth, lower bounded function such that $F(x) \geq F^\star$ for some real value $F^\star \in \mathbb{R}$, then for any $\nu$*

$$\mathbb{E}_\nu\left[\log \pi\right] \leq -2\beta F^\star + \frac{d}{2}\log(\frac{\beta L}{2\pi})$$

*Proof.* We denote by $\Lambda$ the normalization constant of $\pi$ defined as

$$\Lambda = \int_{\mathbb{R}^d} e^{-\beta F(x)}dx.$$

Observe that

$$\log \pi = -\beta F - \log(\Lambda) \leq \beta F^\star - \log(\Lambda)$$

We therefore need only to lower bound $\log(\Lambda)$, which, as performed in Raginsky et al. (2017) Propostion 3.4, can be achieved using a Laplace integral approximation to yield

$$\log \Lambda \geq \beta F^\star + \frac{d}{2}\log(\frac{2\pi}{\beta L})$$

As a consequence, we obtain that

$$\mathbb{E}_{a_t}\left[\log \pi\right] \leq -2\beta F^\star + \frac{d}{2}\log(\frac{\beta L}{2\pi})$$

$\square$

**Lemma 31** (log-Gradient bound). *Let $\pi \propto e^{-\beta F}$ with $F$ an $L$-smooth potential, then for $t \leq \frac{\eta}{\beta}$,*

$$\frac{1}{2}\mathbb{E}_{a_t}\left[\|\nabla \log \pi\|^2\right] \leq 2\beta^2 L^2 \mathcal{W}_2^2\left(a_{\frac{\eta}{\beta}}, \pi\right) + 2\beta L^2 \eta d + Ld$$

*Proof.* Recall from Vempala & Wibisono (2019) Lemma 11 that

$$\mathbb{E}_\pi\left[\|\nabla F\|\right] \leq dL.$$

To control the gradients under a different measure, it suffices to do a simple change of measure as is done in Lemma 12 of Vempala & Wibisono (2019). Since gradients of $F$ are $L$-Lipscthiz, we have that for any $y \in \mathbb{R}^d$ and independent $N' \sim \mathcal{N}(0, I)$

$$\begin{aligned}
\frac{1}{2}\mathbb{E}_{a_t}\left[\|\nabla \log \pi\|^2\right] &\leq \beta^2 \mathbb{E}_{a_t}\left[\|\nabla F(A_t) - \nabla F(y)\|^2\right] + \|\nabla F(y)\|^2 \\
&\leq \beta^2 L^2 \mathbb{E}_{a_t}\left[\|A_t - y\|^2\right] + \|\nabla F(y)\|^2 \\
&\leq 2\beta^2 L^2 \mathbb{E}_{a_t}\left[\|A_t + \sqrt{\frac{\eta}{\beta} - t}N' - y\|^2\right] + 2\beta^2 L^2(\frac{\eta}{\beta} - t)d + \|\nabla F(y)\|^2 \\
&\leq 2\beta^2 L^2 \mathcal{W}_2^2(A_\eta, \pi) + 2\beta L^2 \eta d + Ld,
\end{aligned}$$

where the last inequality is obtained by having $y \sim \pi$ with an optimal coupling. $\square$

### E.1 PROOF OF THE COROLLARY

Here we show how to use our approximate contraction result to obtain a bound on the generalization of SGLD.

**Corollary 20.1** (KL stability under isoperimetry). *Under assumption 19, 20, and assuming $c_\pi \geq 1$. We have for $\eta \leq \frac{\beta}{c_\pi L^2}$, the iterates of equation SGLD $X_k$ and $X_k'$ ran on datasets $\mathcal{D}$ and $\mathcal{D}'$ satisfy*

$$D_{KL}(X_k \| X_k') \leq \frac{poly\left(\frac{\eta}{\beta}, L, d, \sigma, D_{KL}(X_0 \| \pi), D_{KL}(X_0' \| \pi')\right) + C_F + c_\pi^2 S_{Gibbs}}{1 - \gamma} \left(1 - \gamma^{k+1}\right)$$

*where $\gamma = e^{-\eta/4\beta c_\pi}$ and $S_{Gibbs} = \mathbb{E}_\pi\left[\|\nabla F_n(X) - \nabla F_n'(X)\|^2\right]$, and $C_F = \mathbb{E}_{\pi'}\left[\|X\|^2\right] - 2F^\star$.*

*Proof.* Our goal is to apply Theorem 18 to the iterates $X_k$ and $X_k'$. Let us first look at the additive error term erg 8. The additive error term $\mathrm{erg}(a_{\frac{\eta}{\beta}}, b_{\frac{\eta}{\beta}}, \pi, \pi')$ is a sum of Wassertein distances between $a_{\frac{\eta}{\beta}}$ and $\pi$ and $b_{\frac{\eta}{\beta}}$ and $\pi'$. Indeed, we have that

$$\mathrm{erg}(a_{\frac{\eta}{\beta}}, b_{\frac{\eta}{\beta}}, \pi, \pi') = 2\eta\beta^2 L^2 W_2^2(a_{\frac{\eta}{\beta}}, \pi) + (1 + \frac{\eta}{2\beta})W_2^2(a_{\frac{\eta}{\beta}}, b_{\frac{\eta}{\beta}}) + 4W_2^2(b_{\frac{\eta}{\beta}}, \pi')$$

$$\leq 2\eta\beta^2 L^2 W_2^2(a_{\frac{\eta}{\beta}}, \pi) + 2(1 + \frac{\eta}{2\beta})W_2^2(a_{\frac{\eta}{\beta}}, \pi)$$

$$+ 4(1 + \frac{\eta}{2\beta})W_2^2(\pi, \pi') + 8(1 + \frac{\eta}{2\beta})W_2^2(b_{\frac{\eta}{\beta}}, \pi')$$

where we use the triangle inequality for Wassertein distances to obtain the above. We thus have the following three Wassertein distances we need to control: $W_2^2(a_{\frac{\eta}{\beta}}, \pi)$, $W_2^2(b_{\frac{\eta}{\beta}}, \pi')$ and $W_2^2(\pi, \pi')$.

When applied to the iterates of SGLD, the Wassertein distances of interest become $W_2^2(X_{k+1}, \pi)$ and $W_2^2(X_{k+1}', \pi')$ and $W_2^2(\pi, \pi')$. To bound these terms, we therefore need to show that the iterates of SGLD converge in Wassertein distance to their respective target measures $\pi$ and $\pi'$. This was shown in Kinoshita & Suzuki (2022) and we restate their result in Lemma 32. The distances $W_2^2(X_{k+1}, \pi)$ and $W_2^2(X_{k+1}', \pi')$ are thus given by Lemma 32. Now we also know from the log-Sobolev inequality that

$$W_2^2(\pi, \pi') \leq 2c_\pi D_{KL}(\pi \| \pi') \leq c_\pi^2 E_{\pi'}[|\nabla F_n(X) - \nabla F_n'(X)|^2]$$

The first inequality follows from Talagrand's inequality which is implied by the LSI (see 2.2.1 in Vempala & Wibisono (2019). The second inequality is the LSI. We define the stability quantity denoted $S_{Gibbs} := E_{\pi'}[|\nabla F_n(X) - \nabla F_n'(X)|^2]$ to control the right hand side.

Combining the upper bounds given in equation 11 for $X_k$ and $X_k'$, with the above we find that

$$\mathrm{erg}(a_{\frac{\eta}{\beta}}, b_{\frac{\eta}{\beta}}, \pi, \pi') \leq \mathrm{poly}\left(\frac{\eta}{\beta}, L, d, \sigma, D_{KL}(X_0 \| \pi), D_{KL}(X_0' \| \pi')\right) + c_\pi^2 S_{Gibbs}.$$

Adding in the second constant ProbConst 9 and un-rolling the geometric recursion for the iterates $X_k, X_k'$ in Theorem 18 yields the result. $\qquad\square$

**Lemma 32** ($\mathcal{W}_2$ convergence of SGLD Vempala & Wibisono (2019); Kinoshita & Suzuki (2022)). *Under assumptions 19 and 20, for $\eta < \frac{1}{c_\pi L^2}$ the iterates of $X_k$ of SGLD satisfy*

$$W_2^2(X_k, \pi) \leq c_\pi KL(X_0, \pi) + \frac{\eta}{\beta}(8dL^2 + 2\sigma^2) \tag{11}$$

*Proof.* The analysis of Vempala & Wibisono (2019) is sufficient to show this result. As shown in Lemma 3 of Vempala & Wibisono (2019), convergence is established by comparing a single step to the continuous Langevin diffusion with the discrete iterates. Here, in our case, in addition to a discretization error, we further have a stochastic gradient error. In other words, the gap between the drift of the continuous time Langevin diffusion and the discretized SGLD iterates compounds two errors: one for discretization, one for stochasticity. For a continuous Langevin diffusion $(X)_t$ with

drift $\nabla F_n$ started at $X_0$, the expected gap between the continuous Langevin drift and the gradient step of SGLD is given by

$$\mathbb{E}[\|\nabla F_n(X_t) - g(X_0, B)\|_2^2] \leq 2\mathbb{E}[\|\nabla F_n(X_t) - \nabla F(X_0)\|_2^2] + 2\mathbb{E}[\|\nabla F_n(X_0) - g(X_0, B)\|_2^2]$$
$$\leq 2\mathbb{E}[\|\nabla F_n(X_t) - \nabla F_n(X_0)\|_2^2] + 2\sigma^2$$

Consequently, the same analysis as in Vempala & Wibisono (2019) holds, with an additional $2\sigma^2$ term appearing in addition to the discretization error. According to equation 13 in Vempala & Wibisono (2019) it follows that

$$\mathrm{KL}(X_k, \pi) \leq \mathrm{KL}(X_0, \pi) + \frac{\eta(8dL^2 + 2\sigma^2)}{\beta c_\pi}$$

Since $\pi$ verifies the LSI, and the LSI implies Talagrand's T2 inequality Gozlan (2009) the inequality above also gives

$$W_2^2(X_k, \pi) \leq c_\pi \mathrm{KL}(X_0, \pi) + \frac{\eta}{\beta}(8dL^2 + 2\sigma^2).$$

□

# F PROPERTIES OF GAUSSIAN CONVOLUTION

In this section, we prove the two fundamental properties of Gaussian convolution which enable our analysis. We first provide expressions of the log-Hessian, from which both properties of interest follow.

**Lemma 33** (Lemma E.3 of Chen et al. (2022)). *Let $p_\eta = p * \mathcal{N}(0, \eta I)$, we have that*

1. $\nabla^2 \log p_t(x) = Var_{p_{0|\eta}}\left(\frac{Y}{\eta}\right) - \frac{I_d}{\eta}$

2. $\nabla^2 \log p_t(x) = \mathbb{E}_{p_{0|\eta}}\left[\nabla^2 \log p(Y)\right] + Var_{p_{0|\eta}}(\nabla \log p(Y))$

Both the following lemmas follow immediately from the characterization given above. Indeed since Variance terms are p.s.d, we can deduce both the lemmas below. The next Lemma follows from point 1. in 33.

**Lemma 34** (log-Hessian lower bound). *Let $\nu$ be a distribution that results from a Gaussian convolution, i.e, $\nu = \tilde{\nu} \star \mathcal{N}(0, \eta I)$ for some distribution $\tilde{\nu}$, then*

$$\nabla^2 \log \nu \succeq -\frac{1}{\eta} I_d.$$

The next lemma follows from point 2. in 33.

**Lemma 35** (Only upwards). *For any distribution $\nu$ be a distribution, it holds that*
$$\nabla^2 \log \left(\nu \star \mathcal{N}(0, \eta I)\right) \succeq \nabla^2 \log \nu.$$

A direct application of the lemmas allows us to also establish that

**Lemma 36** (Bounded Laplacian). *Let $b$ be a distribution such that $\nabla^2 \log b \succeq -K$, with $K \geq 0$, let $b_t = b \star \mathcal{N}(0, tI)$*
$$\mathbb{E}_{b_t}[\|\nabla \log b_t\|] = -\mathbb{E}_{b_t}[tr\left(\nabla^2 \log b_t\right)] \leq Kd$$

Finally the change of measure lemma is a simple consequence of analysis.

**Lemma 37** (Change of measure). *Let $g : \mathbb{R}^d \to \mathbb{R}$ be a twice-differentiable function such that $\nabla^2 g \succeq -K I_d$ for some $K \in \mathbb{R}$. Then, for any random variables $X, Y$ over $\mathbb{R}^d$, we have*

$$\mathbb{E}[-g(Y)] \leq \mathbb{E}[-g(X)] + \frac{1}{2}\mathbb{E}\left[\|\nabla g(X)\|\right] + \frac{K+1}{2}\mathbb{E}[\|X - Y\|_2^2]$$

*Proof.* From classic results in analysis, we know that

$$g(y) \geq g(x) + \langle \nabla g(x), y - x \rangle - \frac{K}{2}\|x - y\|^2$$

We apply Young's inequality and integrate to obtain the result. □

## G  TECHNICAL LEMMAS

In this section, we include the small technical lemmas that can be found in the literature.

We first show the link between our Lemma 2 and the result of Xu & Raginsky (2017). We first recall their result.

**Lemma 38** (Expected generalization of subgaussian losses, Thm 1 Xu & Raginsky (2017))**.** *Let* $f(w, Z)$ *be a loss function that verifies assumption 1. Then, for any* $k \geq 1$,

$$gen(\mathbb{P}_{X_k|\mathcal{D}}, \mathcal{D}) \leq \sqrt{\frac{2cI(X_k; \mathcal{D})}{n}}.$$

*where* $I$ *denotes the mutual information.*

Let $X_k$ be the output of SGLD ran on a dataset $\mathcal{D}$ with distribution $\mathbb{P}_{X_k|\mathcal{D}}$. Recall that the mutual information is given by

$$I(\mathbb{P}_{X_k|\mathcal{D}}; \mathcal{D}) = \mathrm{D_{KL}}\left((X_k, \mathcal{D}) \| (X_k', \mathcal{D})\right)$$

where $X_k'$ is an independent output of SGLD ran on an independent dataset $\mathcal{D}'$. Conditioning on the second coordinate we have that

$$I(\mathbb{P}_{X_k|\mathcal{D}}; \mathcal{D}) \leq E_D\left[\mathrm{D_{KL}}\left(\mathbb{P}_{X_k|\mathcal{D}} \| \mathbb{P}_{X_k'}\right)\right]$$

since $\mathbb{P}_{X_k'} = \int \mathbb{P}_{X_k'|\mathcal{D}'} \mathbb{P}_{\mathcal{D}'}$, we can invoke Jensen and convexity of the KL divergence to find that

$$I(\mathbb{P}_{X_k|\mathcal{D}}; \mathcal{D}) \leq E_{\mathcal{D}, \mathcal{D}'}\left[\mathrm{D_{KL}}\left(\mathbb{P}_{X_k|\mathcal{D}} \| \mathbb{P}_{X_k'|\mathcal{D}'}\right)\right].$$

Which leads to the KL stability characterization of generalization.

**Lemma 39** (DeBruijn's Identity Zozor & Brossier (2015))**.** *Let* $A$, $B$ *be two random variables over* $\mathbb{R}^d$, *for* $t > 0$, *let* $A_t = A + \sqrt{t}N$ *and* $B = B + \sqrt{t}N$. *Denoting by* $a_t$ *and* $b_t$ *the densities of* $A_t$ *and* $B_t$ *respectively, we have that*

$$\frac{d}{dt}\mathrm{D}_{KL}\left(a_t \| b_t\right) \leq -\frac{1}{2}\mathbb{E}\left[\|\nabla \log a_t - \nabla \log b_t\|^2\right]$$

**Lemma 40** (Integration by parts)**.** *For any two functions* $h, g : \mathbb{R}^d \mapsto \mathbb{R}$

$$\int_{\mathbb{R}^d} \langle \nabla h(x), \nabla g(x)\rangle \, dx = -\int_{\mathbb{R}^d} h(x)\Delta g(x) dx$$

**Lemma 41** (DeBruijn's Identity Zozor & Brossier (2015))**.** *Let* $A$, $B$ *be two random variables over* $\mathbb{R}^d$, *for* $t > 0$, *let* $A_t = A + \sqrt{t}N$ *and* $B = B + \sqrt{t}N$. *Denoting by* $a_t$ *and* $b_t$ *the densities of* $A_t$ *and* $B_t$ *respectively, we have that*

$$\frac{d}{dt}\mathrm{D}_{KL}\left(a_t \| b_t\right) \leq -\frac{1}{2}\mathbb{E}\left[\|\nabla \log a_t - \nabla \log b_t\|^2\right]$$

*Proof.* Since $a_t, b_t$ are undergoing simultaneous heat flow we know that

$$\frac{\partial a_t}{\partial t} = \Delta a_t \qquad \frac{\partial b_t}{\partial t} = \Delta b_t$$

It follows from straightforward computations that

$$\frac{\partial \mathrm{D_{KL}}\left(a_t \| b_t\right)}{\partial t} = \int_{\mathbb{R}^d} \frac{\partial a_t}{\partial t}(x) \log \frac{a_t(x)}{b_t(x)} - \int_{\mathbb{R}^d} \frac{\partial b_t}{\partial t}(x) \frac{a_t(x)}{b_t(x)}$$

$$= -\frac{1}{2}\mathbb{E}_{a_t}\left[\|\nabla \log a_t - \nabla \log b_t\|_2^2\right]$$

$\square$

