# OpenReview forum: "Generalization of noisy SGD under isoperimetry"
_ICLR.cc/2025/Conference — Submitted to ICLR 2025_

### Official Review · Reviewer_DPmq · 2024-10-29

**Soundness:** 2
**Presentation:** 2
**Contribution:** 2
**Rating:** 5
**Confidence:** 4

**Summary:**

This paper addresses stochastic optimization, where the goal is to minimize $F(x):=E_{Z\sim \nu}f(x;Z)$ for some underlying distribution $\nu$.
Let $D$ and $D'$ be two datasets, each consisting of $n$ i.i.d. samples from $\nu$. Running noisy stochastic gradient descent (SGD) on these two datasets yields sequences $\{X_k\}$ and $\{X_k'\}$ respectively. It is known that the generalization error scales with the KL divergence between the distributions of $X_k$ and $X_k'$ .

This paper provides a time-independent upper bound on the KL divergence, even as $k\to \infty$. The authors first show that when the log-Sobolev inequality (LSI) holds, an upper bound on the KL divergence can be derived. They further demonstrate that under appropriate conditions, such as dissipativity, the distribution satisfies the LSI, thus leading to the desired bound.

**Strengths:**

The problem studied is interesting and has practical relevance, with clear motivation provided.
The paper is well-structured, with different cases and scenarios analyzed in depth.
The results are extensively studied across various settings.

**Weaknesses:**

1. The main proofs rely heavily on existing work, like Theorems 6, 8, and 12.
2. Some assumptions require further discussion. For instance, Assumption 15 seems restrictive in unbounded domains like $R^d$.
3. In Theorem 12, the LSI constant scales exponentially with the dimension, which could be problematic for high-dimensional settings.

**Questions:**

1. Isoperimetric vs. Log-Sobolev Inequality:
The paper mentions the use of the isoperimetric inequality, but the arguments seem entirely based on the log-Sobolev inequality (LSI). In probability theory, the isoperimetric inequality is usually considered a separate concept. Could this be a typo or an imprecise reference?

2. What is \Tilde{X}_k' in Theorem 5? Is it a typo, and should it be S_k instead?

3. The $S_k$ is not carefully discussed. In SGLD, when drawing a batch of size $b$, could using a smaller batch size lead to a tighter bound on the KL divergence?

---

> ### Author Response · Authors · 2024-11-20
>
> Dear Reviewer DPmq,
>
> We sincerely thank the reviewer for their time and their review of our paper.
>
> We would like to provide a clarification on the reviewer's summary:
>
> _The authors first show that [LSI] holds, an upper bound on the KL divergence can be derived. They further demonstrate that under  [dissipativity] the distribution satisfies the LSI, thus leading to the desired bound._
>
> Our goal is to show a time-independent generalization bound in non-convex settings. In section 5, we assume the function is dissipative, this allows us to show a per-iterate LSI constant. With this result we are able to prove that noisy SGD applied on an unbounded non-convex loss generalizes (KL-stability)and that it is also differentially private (Renyi stability).
> In section 6, we do not assume dissipativity. We only assume that the target function satisfies the LSI. We derive a technique to establish KL stability without using per-iterate LSI, and only using the final target LSI. This enables us to avoid the poor dimension dependence at the cost of introducing some additional constants in the bound.
>
> Weaknesses:
>
> 1. `Building on prior results`: We would like to stress that theorems 6 and 8 are foundational results. The citation for Theorem 8 for instance refers to Chafai's 2004 book. Theorem 12 which uses Chen et al 2021 result is only applicable _because_ of our preceding analysis. We are building on fundamental results to obtain theorems that are relevant, and our results are not immediate derivations.
> 2. `Assumption 15` For our privacy result, we need a bounded sensitivity assumption because the Renyi divergence introduces expectations that are rather difficult to control. Assumption 15 is standard in the privacy literature (see Def 2.10 in [C]) and it holds for the logistic loss over bounded data or any regularized Lipschitz loss.
> 3. `Dimension dependence`: As we are in a non-convex setting dimension dependence is unavoidable. We note this at the end of section 5 and propose section 6 to improve this dimension dependence.
>
> We answer your questions in the following.
>
> 1. `Isoperimetric vs. Log-Sobolev Inequality`: We followed the terminology introduced in Vempala and Wibisono [B] where functional inequalities like the LSI and the Poincare inequality are referred to as isoperimetric inequalities. This follows from the close equivalences that exist between isoperimetric inequalities and functional inequalities [A]. We will add clarifications.
>
> 3. `What is \Tilde{X}\_k'` Thank you for spotting the typo; there is no tilde. It should simply be $X_k'$ . It is a modified expectation under $X_k'$ (see Definition 21). This is precisely where the difference between KL and Renyi divergences is most seen. For KL, the modified expectation is simple, for Renyi the modified expectation is more complex requiring stronger assumptions like Assumption 15.
> 4. `S_k`: You raise a good point. This quantity measures the gradient difference on two datasets sampled from the same distribution. It is referred to as the 'sensitivity' since it measures how sensitive the model is to changes in the dataset.  As the batch size increases, the gradient estimates approach the population gradient thus making $S_k$ smaller. There is an inverse relationship between the batch size and $S_k$.
>
> We thank you for the time you took to review our work. We remain at your disposal for any further clarifications you may need. If we addressed your concerns we kindly ask that you consider raising your score.
>
> ---
> [A] [Rothaus. "Analytic inequalities, isoperimetric inequalities and logarithmic Sobolev inequalities." *Journal of functional analysis*(1985) ](https://www.sciencedirect.com/science/article/pii/0022123685900795)
>
> [B] [Vempala, Wibisono. "Rapid convergence of the unadjusted langevin algorithm: Isoperimetry suffices." Neurips (2019).](https://proceedings.neurips.cc/paper/2019/hash/65a99bb7a3115fdede20da98b08a370f-Abstract.html)
>
> [C] [Bok, Jinho, Weijie Su, and Jason M. Altschuler. "Shifted Interpolation for Differential Privacy." arXiv preprint arXiv:2403.00278 (2024).](https://arxiv.org/abs/2403.00278)

---

> > ### Comment · Reviewer_DPmq · 2024-11-26
> >
> > Thank you to the authors for their response and for addressing some of my concerns.
> >
> > However, I still have reservations about certain aspects of the results:
> >
> > 1. The proof in Section 4 largely follows from previous work, offering limited novelty.
> > 2. In Section 5, the LSI constant scales exponentially with the dimension, which raises concerns about its practicality.
> > 3. The results in Section 6 are intriguing, particularly because they rely only on the LSI of the stationary distribution while achieving a time-independent KL-divergence bound. However, I find it difficult to appreciate the contributions in Theorem 18 and Corollary 20.1 fully, as the statements are obscured by the use of poly() terms and the ergodicity error term.

---

> ### Author Response · Authors · 2024-11-28
>
> Dear Reviewer DPmq,
>
> We are sincerely grateful that you acknowledged our response. We would like to respond to your remaining reservations in the following.
>
> 1. We had highlighted in our introduction to Section 4 that we first establish `a simplified proof template` on which our contributions in sections 5 and 6 build. Section 4 is setting up the following sections by identifying Theorem 6 as the crucial entrance point for our new results.
> 2. The central goal of our work is to establish a bound that does not diverge as the iteration count increases. Unlike some existing generalization bounds, `our bound is finite and does not scale with iteration counts` $k$ which today exceed $2^{32}$  (see for example training durations in Table 6 of [diffusion models](https://arxiv.org/pdf/2312.02696)). Second, `it is this dimension dependence that we mention is the very reason that motivated our section 6` where we mitigate for this dependence.
> 3. We appreciate that you find our result interesting. We have rewritten the introduction of section 6 in our revision to clarify the proof template. Moreover, `the terms in Theorem 18 are explicitly available in the appendix` (see equations 8 and 9 on line 503). The poly term in the Corollary is merely a combination of line 1153 with Lemma 32.  We restate them here for your convenience, the poly term in Corollary 20.1 is
>
>    $$
>    \left(2\eta\beta^2L^2 + 2(1+\frac{\eta}{2\beta})\right)\left(c_\pi D_\mathrm{KL}(X_0, \pi) + \frac{\eta}{\beta}(8dL^2 +     2\sigma^2)\right) + 8\left(1+\frac{\eta}{2\beta}\right)\left(c_\pi' D_\mathrm{KL}(X_0', \pi') + \frac{\eta}{\beta}(8dL^2 +   2\sigma^2)\right)
>    $$
>    The key of our result is not the exact form of the polynomial dependence above but in the fact that it _is polynomial_. This is   the reason we kept the expansions in the appendix. The takeaway message is the following: at the cost of some additional polynomial terms, we can rely on the LSI of the stationary distribution instead of the per-iterate one. This is the key message (see also [our response here](https://openreview.net/forum?id=0VP3LuzZ8K&noteId=jpjdzSJEsj) for further details).
>
> We thank you for your time and we remain available for further discussions.

---

### Official Review · Reviewer_4Kqp · 2024-11-01

**Soundness:** 3
**Presentation:** 3
**Contribution:** 3
**Rating:** 8
**Confidence:** 3

**Summary:**

This paper studies generalization of stochastic gradient Langevin dynamics (SGLD) via information theoretic bound. The author(s) obtained Renyi stability by assuming the iterates verify the log-Sobolev inequality (LSI). The author(s) further showed that the LSI indeed is satisfied under some dissipativity condition. Further results are obtained when dissipativity is not available, in which case KL stability can still be achieved. The bounds are uniform-in-time which are strong. A by-product is that the paper shows that under dissipativity, all the iterates verify a uniform LSI, which was previously shown only in the strongly-convex setting, that resolves an open question in the literature.

**Strengths:**

(1) The paper is well written, and it is a very solid theoretical paper.

(2) The bounds are uniform-in-time, obtained under Renyi divergence under dissipativity condition and KL stability without dissipativity.

(3) A key ingredient in the proof is to show that under dissipativity, all the iterates verify a uniform LSI, which was previously shown only in the strongly-convex setting. This by-product resolves an open question in the literature.

**Weaknesses:**

(1) Assumption 15 seems to be a really strong assumption. It would be nice if the author(s) can comment on whether this assumption is needed because of the proof technique or it might be unavoidable.

(2) As the author(s) mentioned in the conclusion section, the dimension dependence is strong. But since the author(s) are working with non-convex setting, this is understandable.

**Questions:**

(1) In the second paragraph under contributions, "all the iterates of verify a uniform log-Sobolev inequality''. Should it be "all the iterates verify a uniform log-Sobolev inequality''?

(2) In Lemma 2, is the constant $c$ the same one from Assumption 1? If so, you should mention in the statement of Lemma 2 that you are assuming Assumption 1. For a related question, for each lemma and theorem, it would be really nice if the author(s) can make it more transparent which assumptions are used, especially because the paper contains quite many theoretical results in different settings which require different assumptions.

(3) It would be nice if the author(s) can add some intuitive explanations about the half-step technique in the analysis. For example, when you split the Gaussian noise $N_{k+1}$ into $N_{k+1}^{(1)}$ and $N_{k+1}^{(2)}$, why the former becomes expansive, whereas the latter becomes contractive.

(4) Assumption 14 seems to be a bit strange. If it is pseudo-Lipschitz, shouldn't it be small
when $z$ and $z'$ are close to each other but I do not see $z$ and $z'$ appearing on the right hand side.

---

> ### Author Response · Authors · 2024-11-20
>
> Dear Reviewer 4Kqp,
>
> We are grateful for your thorough review and feedback. We are encouraged by the strengths you see in our work. In the following, we would like to answer some of your questions.
>
> (W1) `Assumption 15` For our privacy result, we need a bounded sensitivity assumption because the Renyi divergence introduces expectations that are rather difficult to control. Assumption 15 is standard in the privacy literature (see Def 2.10 in [A]) and it holds for the logistic loss over bounded data or any regularized Lipschitz loss.
>
> Questions:
>
> (1) `In the second paragraph under contributions.` Thank you for spotting this typo.
>
> (2) `In Lemma 2, is the constant the same one from Assumption 1?` Yes the constant is the same and we have fixed the theorem statement.
>
> (3) `Some intuitive explanations about the half-step technique`: The expansion is done by the gradient step and the contraction is done by the noise. Conditioned on the current step, the KL divergence measuring the gradient step alone would be infinity as it would correspond to the KL divergence between two Dirac distributions. However, if we split the noise in two and add some Gaussian noise to the gradient step, the KL divergence becomes finite and easy to analyze. This is the main reason why we split the noise. A secondary reason is the application of Lemma 17: we need a small amount of Gaussian noise to smoothe the distribution before analyzing the contraction.
>
> (4) `Assumption 14`. Indeed you are correct, we borrow the assumption from prior work [B, Assumption 3.1] where there is a $\|z - z'\|$ appearing on the RHS. We then absorbed $\|z - z'\|$ into $\theta$ and $D$ by assuming a bounded data distribution. We will add clarifications.
>
> We thank you for the time you took to review our paper and remain at your disposal for any clarifications you may need.
>
> References
> ---
> [A] [Bok, Jinho, Weijie Su, and Jason M. Altschuler. "Shifted Interpolation for Differential Privacy." arXiv preprint arXiv:2403.00278 (2024).](https://arxiv.org/abs/2403.00278)
>
> [B] [Zhu, Lingjiong, et al. "Uniform-in-time Wasserstein stability bounds for (noisy) stochastic gradient descent." Advances in Neural Information Processing Systems 36 (2024).](https://proceedings.neurips.cc/paper_files/paper/2023/hash/05d6b5b6901fb57d2c287e1d3ce6d63c-Abstract-Conference.html)

---

### Official Review · Reviewer_aiX5 · 2024-11-03

**Soundness:** 4
**Presentation:** 3
**Contribution:** 2
**Rating:** 6
**Confidence:** 4

**Summary:**

The paper explores KL and Rényi divergence stability of Stochastic Gradient Langevin Dynamics (SGLD) algorithm. The main characteristic of the presented stability bounds is that they do not become vacuous with the number of iteration of SGLD, which is achieved by assuming log-Sobolev type isoperimetric inequality being satisfied, either throughout the stochastic process, or just by the steady-state Gibbs distribution that SGLD asymptotically approximates. Such isoperimetric properties have also been recently shown to provide rapid convergence in informational divergence as well as convergent DP properties. In a similar vein, the paper derives non-asymptotic and convergent generalization bounds for SGLD as well as bounds on Rényi DP under isoperimetric assumptions. Moreover, the paper shows that the isoperimetric assumption is satisfied under settings considerably milder than strongly-convex losses, such as under dissipative and smooth losses.

**Strengths:**

- The paper is well-written and explains its ideas in a well-paced manner, introducing a bit of background, assumptions, and supporting theorems at convenient locations for the reader to follow.
- The paper presents an example where non-convex loss can provide generalization guarantees that are non-vacuous in number of iterations.
- The paper simplifies the expansive-contractive decomposition of SGLD steps used in related works for bounding information divergence.

**Weaknesses:**

Non-analytical issues:

- There is no comparison of the presented bounds with existing results in literature. The generalization bounds presented should be compared to both information-theoretic and non-information theoretic bounds under similar sets of assumptions.

- Contrasting with lower bounds on stability is needed to assess gaps in the tightness of the analysis presented. If such an analysis proves difficult, a well-designed experimental evaluation to compare the generalization bounds with the actual generalization behaviour under the stated assumptions should have been included.

Technical comments:

- Firstly, information-theoretic generalization bounds inspired by Xu and Raginsky seem to have an O(1/sqrt(n)) dependence on the dataset size, even in cases where other generalization approaches give a better O(1/n) bounds [1]. Since the bounds presented in this paper show dependence in the dataset size n only through Lemma 2 (by Xu and Raginsky), I believe the paper's generalization guarantees might have suboptimal dependence on n under the assumptions made.

- Lemma 3 for conversion of Rényi DP to $(\epsilon, \delta)$-DP isn't the best known bound. [Theorem 21, 2] gives a strict improvement which is the best known conversion in my knowledge.

- While Theorem 7 neatly presents the change in Rényi divergence under LSI after a single SGLD step, I believe this inequality might be loose, specially in the dependence on the order $q$ of Rényi divergence. That's because the paper slightly modifies the expansion-contraction template used in other prior works for simplicity. In [3] the expansion-contraction step seems to occur simultaneously, which yield a PDE that is better able to quantify the change in Rényi divergence when integrated over a single step.

- In Section 5.1, the constant of LSI under convexity is dimension independent. But on relaxing strong convexity to dissipativity, the LSI constant has an exponential dependence O(e^d) on the dimension size. The paper further claims in line 418 that this dependence on dimension can't be improved without additional assumptions. To me, this seems like a major hurdle that greatly limits the applicability of the generalization bounds presented (both Corollary 14.1 and 15.1) as plugging in the $C_{LSI}$ constant of Theorem 12 gives an $KL(X_t\Vert X'_t) = O(e^d)$ dependence on dimension $d$.


[1] Haghifam, Mahdi, et al. "Limitations of information-theoretic generalization bounds for gradient descent methods in stochastic convex optimization." International Conference on Algorithmic Learning Theory. PMLR, 2023.

[2] Balle, Borja, et al. "Hypothesis testing interpretations and renyi differential privacy." International Conference on Artificial Intelligence and Statistics. PMLR, 2020.

[3] Chourasia, Rishav, Jiayuan Ye, and Reza Shokri. "Differential privacy dynamics of langevin diffusion and noisy gradient descent." Advances in Neural Information Processing Systems 34 (2021): 14771-14781.

Crafting examples of loss functions satisfying the assumptions made and computing a lower bound on how the KL or Rényi divergence changes with iterations seems doable.

**Questions:**

I'm not sure if I understood the results in Section 6, which seems to be adopting an entirely different style of analysis as compared to section 5, which helps in lifting the LSI assumption on the entire sequence of intermediate distributions to LSI on the Gibbs distribution corresponding to the loss function. It would help if this approach is explained more thoroughly to see the idea in there a bit more clearly.

I'm open to increasing my score, especially if Section 6 has some good ideas that I might have missed.

---

> ### Author Response · Authors · 2024-11-20
>
> Dear Reviewer aiX5,
>
> We are grateful for your thorough, well organized review and your detailed feedback. In the following, we would like to address the points you raise.
>
> 1. `bounds in the literature` To strengthen our Related Work Section 3,  we will add a table listing previous bounds considering iterative noisy gradient schemes in non-convex settings. Our focus in our paper is on bounds that apply specifically to noisy iterative algorithms and not on bounds that are agnostic to the learning algorithm, as is the case for most generalization bounds which only consider the function class.
>
> 2. `lower bounds` Our motivation is to amend a gap in previous bounds which explode as the iteration count increases. Our dissipative bound, unlike previous results, matches the lower bound of (Chourasia et al 2021, Theorem 3), for $q=1$,  as strongly convex functions are special cases of dissipative functions. Our bounds in section 6, however, are unlikely to be tight but they are a significant improvement over previous diverging bounds.
>
>  3. `experimental evaluation` We believe we are in a reverse situation where practice has already shown that large training times can perform well (see for instance the $2^{32}$ training iterations of state of the art [diffusion models](https://arxiv.org/pdf/2312.02696)) and generalization bounds that aligned with these observations were absent as most existing bounds diverged with training time. Our work is amending theory to make it match what is already practically observed. Our motivation is to show, theoretically, that there exist non-convex settings where long training runs are not harmful.
>
> Technical comments:
>
> 1. `O(1/sqrt(n)) vs O(1/n) bounds` For information-theoretic bounds the fast rate is indeed achievable but it comes with a trade-off: it becomes analytically difficult to obtain a decay factor to control the number of iterations. With current techniques, to obtain the fast rate one has to accept a diverging bound, see Proposition 15 and Corollary 16 of [D]. Our goal is to characterize algorithms that are run for a large number of iterations. For this reason, we cannot use $1/n$ generalization bounds that only depend on the function space nor can we use information-theoretic bounds which are not tractable when applied to noisy iterative algorithms.
>
> 2. `Conversion of Rényi DP to DP` We are very grateful for this reference, we were unaware of this refinement. We will include the improved conversion in Lemma 3.
>
> 3. `the dependence on the order of Rényi divergence` For the privacy result, indeed, your observation that the decay factor in (Chourasia et al 2021) does not appear to degrade with $q$ is correct. The reason is not the two-step analysis but rather the tighter "Renyi log-Sobolev inequality" they use. Their Lemma 3 has a $\partial R_\alpha/\partial\alpha$ appearing which is the derivative of the Renyi divergence with respect to the order. As Renyi divergence is increasing, this term is positive and can be ignored. By disregarding it, we obtain our result. When including it in the analysis, Chourasia et al obtain the equivalent of our Theorem 6 *but with a changing Renyi order per iteration* (see their equation 66-67). We chose not to operate with changing Renyi orders as it quickly becomes notationally heavy but their tightening can be applied to our result with a similar PDE argument (their lines 59-61).
>
>
>
> 4. `dimension dependence` Going from the strongly convex setting section 5.1 to the non-convex setting incurs a dimension dependence. This worst-case dimension dependence in noisy iterative schemes is expected in non-convex settings (see discussion around 4.3 in [B]). However, we *mitigate for this poor dimension dependence in Section 6*.
>
> 5. `a lower bound on how the KL or Rényi divergence changes with iterations seems doable` Lower bounds in non-convex settings are notoriously difficult to construct. In strongly convex settings, the Gaussian serves to establish lower bounds, in the non-convex setting, the KL divergence between even the simplest non-convex distribution, a mixture of Gaussians, is not analytically tractable. In the space of noisy iterative algorithms, lower bounds are a major open problem (see page 3 of [C]).

---

> > ### Author Response · Authors · 2024-11-20
> >
> > `Ideas in section 6`
> >
> > In section 5, we use a per-iterate-LSI to obtain a decaying factor. When all we have is the dissipative assumption, the per-iterate-LSI is set by the worst-case dissipative function. This is where the worst-case exponential dependence enters. This is the cost we must pay to have a bound where each constant involved relates to stability and it is the cost of matching the strongly convex bound of (Chourasia et al 2021).
> >
> > In section 6, we accept other constants in the bound namely, the convergence speed of the algorithm. With this allowance, we can use the LSI of the target instead of a per-iterate LSI. Here the constants are no longer set by the worst dissipative function, but by the specific $F_n$ we are optimizing. For this benefit, we must introduce terms polynomial in dimension in our bounds, we also need the algorithms to converge in Wassertein 2, which sets the stepsize in Corollary 20.1.
> >
> > `Techniques in Section 6`:
> >
> > The main technical tool is Theorem 18. There we show an approximate contraction that can replace Theorem 6 which requires a per-iterate-LSI. To obtain this approximate contraction, explained roughly, we must change the second argument in the divergence (the b in KL(a||b)) from the distribution of the iterates to the distribution of the target. This change of measure is not generally possible.
> >
> > The simplification of the expansion-contraction plays a significant role and will allow us to perform the change of measure. We need the half-noises to play two roles. The first noise plays a smoothing role. It guarantees that the hessian is lower bounded. This smoothness property allows us to do a change of measure in Lemma 18, and instead of the iterate LSI, we can use the LSI of the target.
> >
> > To summarize, section 5 shows
> > *  A per-iterate-LSI. This is a valuable contribution as it solves a curious quirk: it was unknown for example if discrete langevin iterates initialized at a Gaussian targetting a mixture of two Gaussians had a bounded LSI constant when both the initialization and the target had a finite one.
> > * a time-independent bound which matches the form of the strongly convex bound.
> >
> > Then, because of the dimension dependence of the bound, we propose section 6 where
> > * A dependence on the target LSI instead of a per-iterate-LSI is achieved at the cost of introducing constants that do not appear in the strongly convex case.
> > * A change of measure argument is shown thanks to the half-step technique. Fundamentally it relies on introducing well-chosen couplings on the RHS of Lemma 18 to make Wassertein distances appear.
> >
> >
> > We are grateful for the time you took evaluating our work, we remain at your disposal for any further clarifications we could provide. If we have addressed your concerns we kindly ask you to consider raising your score.
> >
> > References
> > ---
> > [B] [Raginsky, Rakhlin, Telgarsky. "Non-convex learning via stochastic gradient langevin dynamics: a nonasymptotic analysis." Conference on Learning Theory. PMLR, 2017](https://arxiv.org/abs/1702.03849).
> >
> > [C] [Chewi, Sinho, et al. "Fisher information lower bounds for sampling." arXiv preprint arXiv:2210.02482 (2022).](https://arxiv.org/abs/2210.02482)
> >
> > [D] [Wang, Hao, Rui Gao, and Flavio P. Calmon. "Generalization bounds for noisy iterative algorithms using properties of additive noise channels." Journal of machine learning research 24.26 (2023): 1-43.](https://www.jmlr.org/papers/v24/21-1396.html)

---

> ### Comment · Reviewer_aiX5 · 2024-11-25
>
> Most of the authors comments are well received. I have raised my score accordingly. Here are the concerns that weren't sufficiently addressed.
>
> > "For this reason, we cannot use generalization bounds that only depend on the function space nor can we use information-theoretic bounds which are not tractable when applied to noisy iterative algorithms."
>
> Authors goal for the paper is to "characterize algorithms that are run for a large number of iterations" and the generalization bound presented has the desired $O(1)$ dependence on number of iteration. But since the dependence on the dataset size $n$ suffers (perhaps due to fundamental limitation of the analytical tools used, perhaps not), I believe it is important to acknowledge this gap in the paper.
>
> > "Our dissipative bound, unlike previous results, matches the lower bound of (Chourasia et al 2021, Theorem 3), for
> , as strongly convex functions are special cases of dissipative functions."
>
> It seems to me like this is an incorrect assertion. The LSI constant in Theorem 12 hides a dependence on dimension d that does not appear in the lower bound lower bound of (Chourasia et al 2021, Theorem 3). The paper should acknowledge that the bounds presented under dissipativity might not be tight (at least in dimension d and in dataset size n).

---

> > ### Author Response · Authors · 2024-11-28
> >
> > Dear Reviewer aiX5,
> >
> > We thank you for your comments and we have updated our manuscript according to your feedback. In particular, we have added a discussion in Appendix.A on the fast rate of $1/n$ and mentioned that it is an interesting open problem to both have no dependence on the iteration count _and_ have the optimal dependence on $n$.
> >
> > With regards to the LSI constant in section 5:  The worst dissipative function, unlike the worst strongly convex function, does have an exponentially bad LSI constant. The dependence on $d$ is thus necessary and is due to the larger, non-convex, class of functions considered (see the hierarchy in Figure 2) and not a looseness in the analysis. We use the term "match" to refer to the fact that no additional terms are introduced. *Section 6 luckily does not depend on the worst dissipative function, but rather on the currently optimized function*.
> >
> > We will add a clear sentence remarking on $n, d$ and the potential for future new techniques to tighten the bounds. We are again grateful for your time, and we remain available for any further clarifications we may provide.

---

### Official Review · Reviewer_J6MA · 2024-11-04

**Soundness:** 3
**Presentation:** 2
**Contribution:** 3
**Rating:** 6
**Confidence:** 2

**Summary:**

This paper studies the stability of SGLD, which implies generalization and differential privacy guarantees of SGLD. Instead of assuming strong convexity of the loss function, the authors demonstrate that stability results still hold under the dissipativity assumption. Technically, their result is established via verify the uniform LSI of SGLD outputs. Beyond the dissipativity assumption, they also establish a stability result via utilizing the regularizing properties of Gaussian convolution.

**Strengths:**

1. Clear statement of setup and theoretical results.

2. Detailed proof with several illustrations of proof steps via pictures.

3. Previous results are clearly mentioned with detailed references.

4. The results in this paper extend previous findings under convexity to weaker conditions, which is an important improvement.

**Weaknesses:**

The writing in some parts is confusing, making it difficult to clearly understand the contribution in Section 6:

1. In line 199, the authors state, "we assume in the following that the Gibbs distribution with density proportional to exp(-Fn) satisfies the LS," but in Assumption 19, the authors seem to state this LSI assumption again. Is there any difference?

2. In Section 6.1, the authors seem to claim two important preliminary results in Lemma 16 and 17 but don't explain how they affect establishing the main result.

3. It seems that the results in Section 6 are established without verifying the uniform LSI. If so,I am wondering if the analysis template in Section 4 is only applied in Section 5 and whether it should be merged with Section 5. Moreover what is the main proof framework for establishing results in Section 6?


Other minor writing problems

1. In line 90, should it be "the bound does not decay to zero"?

2. In lines 439, 452, 874, "given in Theorem 12."

3. In line 504, "given in equation 8 and equation 9."

**Questions:**

My main questions are about Section 6, as stated in the weaknesses part.

---

> ### Author Response · Authors · 2024-11-20
>
> Dear Reviewer J6MA,
>
> We are grateful for your thorough review and your feedback. We would like to address the points you have raised.
>
>
> W. 1. `Assumption 19 and LSI` They are the same assumption. We made a typographic choice on our part to have theorems preceded by the assumptions they use. It is for this reason that we restate the LSI assumption formally in Assumption 19. Note that dissipativity is a stronger assumption than Assumption 19 (see Figure 2). So in our analysis, Section 5 operates with the stronger dissipativity assumption and Section 6 operates under the relaxed Assumption 19.
>
> W. 2&3. `Section 6`. We hear the reviewer's feedback and we have rewritten the introduction of Section 6 to make our contribution clear. Both sections 5 and 6 rely on the template in section 4. Indeed, A central component in the analysis is the step-wise contraction with a factor $\gamma$ in Theorem 6. The existence of a contraction factor is what ensures a finite bound. In section 5, we analyze dissipative functions to show that a uniform LSI exists which gives the desired $\gamma$ decay factor. In Section 6, we notice that instead of a clean contraction in Theorem 6, we can instead have
> $$
> D(current) \leq \gamma D(previous) + \textrm{additive terms}
> $$
> where the additive terms are bounded. This is what we call approximate contraction: Theorem 18 has a similar contraction as in Theorem 6 but with added additive terms. By substituting Theorem 6 with Theorem 18, we obtain our KL generalization result.
>
> `what is the main proof framework`. The main technical contribution in section 6 lies in showing Theorem 18. Instead of taking a per-iterate LSI like in theorem 6, Theorem 18 only requires LSI of the target measure. To obtain this result, one needs to change, at some point in the analysis, the distribution with respect to which the KL divergence is being taken: from the distribution of the iterates to the distribution of the the target. To perform this change of distribution, we use Lemma 18. Observe there that an expectation under $Y$ on the left can be switched to an expectation under $X$ on the right. Lemma 18 however only applies to sufficiently smooth functions. This is precisely where Lemma 17 comes in. Lemma 17 shows that Gaussian convolution has a smoothing behavior which makes us able to apply Lemma 18. Combining these results with a continuous time analysis of Gaussian convolution yields Theorem 18.
>
>
>
> `Writing lines 90, 439, 504`: Thank you for spotting these mistakes.
>
> We are grateful for your time and remain available for any further clarifications we could provide.

---

### Meta-Review · Area_Chair_HERJ · 2024-12-20

**Metareview:**

The authors studied the generalization properties of SGLD under isoperimetry. From what I understand, the results essentially generalizes the results of Raginsky et al. (2017) to a KL based stability bound.

As many of the reviewers would agree, this line of work has been around for a while, and there is not a lot of significant contributions towards improving the critical issues here. In particular, any isoperimetric approach in a non-convex setting suffers from the curse of dimensionality, i.e. the constants depend exponentially on dimension. This is a critical issue as we will never be able to use these bounds in practice.

Let's put aside this fundamental issue for a bit, most of the ingredients in this paper were not new to the reviewers or myself, nor were the results significant improvements. The claim of resolving the open question of Vempala and Wibisono (2019) is also overstated, as you require a stronger dissipativity assumption, and the uniform LSI constant is exponentially dependent on dimension once again.

Given that many reviewers believing the contributions are lacking and the above discussion, I will recommend reject.

**Additional Comments On Reviewer Discussion:**

Reviewer aiX5's discussion was the most productive, as it highlighted many of the issues I discussed above that remained unresolved. This line of messages was the most informative towards my decisions.

---

### Decision · Program_Chairs · 2025-01-22

Reject